# Enhancing Stability of Physics-Informed Neural Network Training Through Saddle-Point Reformulation

**Dmitry Bylinkin**[1,2]      **Mikhail Aleksandrov**[1,2,3]      **Savelii Chezhegov**[1,2,4]

**Aleksandr Beznosikov**[1,2,5]

[1]Basic Research of Artificial Intelligence Laboratory (BRAIn Lab)
[2]Moscow Independent Research Institute of Artificial Intelligence
[3]Lomonosov Moscow State University
[4]Sber AI Lab
[5]Innopolis University

## Abstract

Physics-informed neural networks (*PINN*s) have gained prominence in recent years and are now effectively used in a number of applications. However, their performance remains unstable due to the complex landscape of the loss function. To address this issue, we reformulate *PINN* training as a nonconvex-strongly concave saddle-point problem. After establishing the theoretical foundation for this approach, we conduct an extensive experimental study, evaluating its effectiveness across various tasks and architectures. Our results demonstrate that the proposed method outperforms the current state-of-the-art techniques.

## 1 Introduction

Mathematical physics is a cornerstone of modern science. It provides powerful tools for theoretical studies and finds applications in practical fields. One of its central challenges is solving partial differential equations (PDEs) (Bateman, 1932; Evans, 2022). They arise in the formal description of phenomena ranging from heat diffusion to quantum mechanics and typically take the form of a boundary value problem involving differential operators on some domain (Yakubov and Yakubov, 1999). Generally, there is a system of $M_r$ equations and $M - M_r$ boundary/initial conditions:

$$\begin{aligned} \mathcal{R}_i[u](x) = f_i(x), \ i \in [1, M_r], \ x \in \Omega; \\ \mathcal{B}_j[u](x) = g_j(x), \ j \in [M_r + 1, M], \ x \in \partial\Omega, \end{aligned} \tag{1}$$

where $f_i, g_i : \mathbb{R}^d \to \mathbb{R}$ are the scalar functions; $\mathcal{R}_i[u], \mathcal{B}_i[u] : \mathbb{R}^d \to \mathbb{R}$ are the operators actions on the mapping $u : \mathbb{R}^d \to \mathbb{R}^m$; $\Omega \subset \mathbb{R}^d$ and $\partial\Omega \subset \mathbb{R}^{d-1}$ are the domain set and its boundary, respectively. Since exact solutions are rare outside idealized cases, the community is focused on developing numerical methods. Among the most established techniques are those based on finite differences (Courant et al., 1967), volumes (Patankar and Spalding, 1983), and elements (Courant et al., 1994). Despite high accuracy and computational efficiency of traditional approaches, they require substantial time to interpolate a new solution (Grossmann et al., 2024, Figures 4b,6b), (Liu et al., 2024b, Figure 6d-f). This limitation makes them impractical in problems where runtime is the primary performance metric. A promising direction for addressing this issue lies in machine learning, due to the low inference time of small neural networks (Guo et al., 2016; Zhu and Zabaras, 2018; Yu et al., 2018). Although the concept of approximating the solution with a parametrized function $u(\theta)$ is quite old and dates back to the works of Meade Jr and Fernandez (1994); Dissanayake and Phan-Thien (1994); Lagaris et al. (1998), it has only recently gained attention under the name *PINN (physics-informed neural network)* (Raissi et al., 2019). While initial results in this area were obtained using *MLP*s, advanced architectures such as learned activations (Jagtap et al., 2020a;b), memory (Krishnapriyan et al., 2021; Cho et al., 2023; Liu et al., 2024b) and attention (Zhao et al., 2023; Anagnostopoulos et al., 2024) have led to significant improvements. *PINN*s are trained via

the empirical risk minimization (ERM) (Raissi et al., 2019):

$$\min_{\theta \in \mathbb{R}^d} \left[ \mathcal{L}(\theta) = \sum_{i=1}^{M_r} \mathcal{L}_{r,i}(\theta) + \sum_{j=M_r+1}^{M} \mathcal{L}_{b,j}(\theta) \right], \text{ with } \mathcal{L}_{r,i}(\theta) = \frac{1}{N_r} \sum_{n=1}^{N_r} \left[ \mathcal{R}_i[u(\theta)](x_r^n) - f(x_r^n) \right]^2,$$

$$\mathcal{L}_{b,j}(\theta) = \frac{1}{N_b} \sum_{n=1}^{N_b} \left[ \mathcal{B}_j[u(\theta)](x_b^n) - g(x_b^n) \right]^2,$$

where $\{x_r^n\}_{n=1}^{N_r}$, $\{x_b^n\}_{n=1}^{N_b}$ are the sets of samples belonging to the interior and boundary of $\Omega$, respectively; $N_r$, $N_b$ are the sizes of corresponding datasets.

While achieving speed up compared to traditional approaches, *PINN*s face significant challenges during the training phase. There is no guarantee that $\arg\min_{\theta \in \mathbb{R}^d} \mathcal{L}(\theta)$ minimizes every $\mathcal{L}_{r,i}(\theta)$ and $\mathcal{L}_{b,j}(\theta)$ individually. In practice, their corresponding gradients $\nabla \mathcal{L}_{r,i}(\theta), \nabla \mathcal{L}_{b,j}(\theta)$ have highly dissimilar magnitudes (Hwang and Lim, 2024, Figure 2). Consequently, the parameter updates become biased toward one of the loss terms, and the solution is approximated only on the boundary or only inside the domain (Hwang and Lim, 2024, Figure 1). Most successful approaches employ weights $\pi = (\pi_1, \ldots, \pi_M)^\top$ selected from the set $S$, typically the unit simplex, to balance competing losses for $\mathcal{R}_i[u]$, $\mathcal{B}_j[u]$ (Wang et al., 2021; Jin et al., 2021; Wang et al., 2022; Son et al., 2023; Hwang and Lim, 2024). The situation is further complicated by the fact that optimization landscape substantially varies over physical problems (Krishnapriyan et al., 2021, Figure 3). Therefore, a scheme that performs well for one PDE may turn out to be inadequate for another (Hao et al., 2023, Table 3), and selecting an appropriate optimizer requires case-by-case search.

In our work, we consider training *PINN* as a non-Euclidean saddle-point problem (SPP):

$$\min_{\theta \in \mathbb{R}^d} \max_{\pi \in S} \left[ \mathcal{L}(\theta, \pi) \right], \text{ with } \sum_{i=1}^{M_r} \pi_i \mathcal{L}_{r,i}(\theta) + \sum_{j=M_r+1}^{M} \pi_j \mathcal{L}_{b,j}(\theta) - \lambda D_\psi(\pi || \hat{\pi}), \tag{2}$$

where $D_\psi(\cdot || \hat{\pi})$ is the Bregman divergence (Nemirovskij and Yudin, 1983). Thus, if some operator is underestimated relative to another one, its weight is increased, as does its contribution to the loss function. We also introduce the hyperparameter $\lambda$ to enable control over the weights via the penalty for deviating from the reference distribution $\hat{\pi}$, typically the uniform one. A similar methodology was considered in (Liu and Wang, 2021). However, the authors provided no theory and examined the Euclidean case, which may be unsuitable if $S$ has a complex geometry. For example, if $S$ is a unit simplex, then KL-divergence is the preferable distance measure, particularly because it accounts for relative rather than absolute changes in weights.

There is currently no guaranties for the nonconvex problem (2), and this setting remains empirically underexplored for *PINN*s. In this work, we overcome both theoretical and practical challenges to investigate the feasibility of training physics-informed neural networks as non-Euclidean SPPs.

## 2 RELATED WORKS

### 2.1 LOSS RESCALING IN GENERAL CASE

Training a physics-informed neural network is a special case of multi-task learning (Zhang and Yang, 2021). Various rescaling techniques had been developed in this domain by the time of the emergence of *PINN*s. Chen et al. (2018) suggested treating weights as trainable functions $\pi_m(\hat{\theta})$. They defined a separate loss such that the norm of a single task gradient $\nabla(\pi_m(\hat{\theta}) \mathcal{L}_{\cdot,i}(\theta))$ is close to the sum of the other gradients. A similar approach was explored in (Kendall et al., 2018). However, using neural networks to evaluate the parameters leads to increased memory consumption. As a consequence, the community has developed a number of computationally less expensive techniques. Sener and Koltun (2018) proposed solving a quadratic optimization problem on a unit simplex to determine $\{\pi_m\}_{m=1}^{M}$. Furthermore, approaches that calculate weights via zero- and first-order statistics have gained attention due to their combination of efficiency and quality (Liu et al., 2019; Yu et al., 2020; Heydari et al., 2019; Chen et al., 2018; Wang et al., 2020).

### 2.2 LOSS RESCALING IN *PINN*S

The unique challenges posed by PDEs and physical constraints motivated the development of weighting techniques specifically for *PINN*s. Wang et al. (2021) were among the first in this di-

rection. Inspired by ideas behind `Adam` (Kingma and Ba, 2014), they proposed a learning rate annealing procedure that automatically tunes $\{\pi_m\}_{m=1}^M$ by utilizing the back-propagated gradient statistics. To mitigate the high variance inherent in the stochastic nature of updates, the authors suggested computing the actual weights as a running average of their previous values. This scheme was then understood in greater depth (Jin et al., 2021; Maddu et al., 2022; Bischof and Kraus, 2025). As an orthogonal approach, in (Wang et al., 2022), loss rescaling was addressed from a neural tangent kernel perspective. Despite the advances, it may be computationally expensive. Indeed, the use of the Jacobian poses a challenge when solving nonlinear equations, as it is not constant in that case (Bonfanti et al., 2024). In parallel to these commonly used approaches, a number of exotic non-benchmarked techniques exist. For example, schemes based on likelihood (Xiang et al., 2022; Hou et al., 2023), augmented Lagrangian (Son et al., 2023) and conjugate cone (Hwang and Lim, 2024).

## 2.3 NONCONVEX-STRONGLY CONCAVE SPPs

The theory of SPPs is constructed mostly for convex-concave objectives (Korpelevich, 1976; Nemirovski, 2004; Du and Hu, 2019; Adolphs et al., 2019; Beznosikov et al., 2023). However, the problem (2) falls outside of this class, since the complex nature of differential operators implies a poor non-convex landscape in $\theta$. On the other hand, in terms of the weights $\pi$, $\mathcal{L}(\theta, \pi)$ is guaranteed to be strongly concave regardless of the PDE being solved. Nonconvex-concave (N-C) and nonconvex-strongly concave (N-SC) SPPs remain poorly understood. Today's research focuses on modifying two-timescale gradient descent-ascent (`TT-GDA`), which has demonstrated success in training GANs (Heusel et al., 2017). Using a double-loop scheme, Nouiehed et al. (2019) achieved an $\varepsilon$-solution in $\tilde{\mathcal{O}}\left(\kappa^4/\varepsilon^2\right)$ iterations, where $\kappa$ denotes the condition number of the objective in the concave component. Assuming `max-oracle` to be available, Jin et al. (2019) improved this result to $\tilde{\mathcal{O}}\left(\kappa^2/\varepsilon^2\right)$. Several triple-loop techniques were developed in (Thekumparampil et al., 2019; Kong and Monteiro, 2021). However, nested loops are challenging to implement and tune in practice. At the same time, providing a theoretical analysis directly to `TT-GDA` posed a challenge. This was finally done in (Lin et al., 2020) with a complexity of $\mathcal{O}\left(\kappa^2/\varepsilon^2\right)$. Later, the result was generalized by Xu et al. (2023). They provided unified analysis of single-loop schemes for N-C problems.

A key drawback of the mentioned methods is the use of Euclidean setting. This may be inappropriate for describing the geometry of $S$ in the problem (2) (Mohri et al., 2019; Mehta et al., 2024). Consequently, there is an interest in searching for alternatives. Huang et al. (2021) considered a setup that is non-Euclidean in the non-convex component and Euclidean in the strongly concave one. However, in our paper, we need the opposite. Indeed, in the problem (2), $\theta$ lies in $\mathbb{R}^d$ and is therefore suited to the Euclidean distance, while $\pi$ demands a more complicated description. Thus, this work is not suitable for our purposes, although it provides useful intuition. Boroun et al. (2023) employed `Frank-Wolfe` (Jaggi, 2013) to perform both ascent and descent steps. However, exploiting non-regularized linear approximation yields sparse values of $\{\pi_m\}_{m=1}^M$, which may result in unstable convergence.

## 3 OUR CONTRIBUTION

Currently, each PDE has its own dominant optimization method: `LRA` (Wang et al., 2021) for *Poisson*, `RAR` (Lu et al., 2021) for *Heat*, `NTK` (Wang et al., 2022) for *Wave*, and `Adam` (Kingma and Ba, 2014) for *Navier-Stokes*. We study the potential of minimizing the *PINN*'s objective via the non-Euclidean saddle-point problem (2) in order to make the training porcess robust. The paper presents a comprehensive theoretical and empirical analysis of this approach.

Table 1: Comparison of SOTA results. Mean **L2RE** over 3 runs is used as a quality metric.

| Approach | Poisson | Heat | Navier-Stokes | Wave | High dim |
|---|---|---|---|---|---|
| Prev. best | 1.14E-2 | 1.74E-2 | 4.67E-2 | 9.20E-2 | 4.69E-4 |
| This paper | **8.15E-3** | **1.40E-2** | **2.35E-2** | **1.63E-2** | **1.31E-4** |

• **Theoretical foundation.** Studying nonconvex-strongly concave SPPs with non-Euclidean geometry of the strongly concave component, we propose a method based on a suitable Bregman proximal mapping. We develop a rigorous theory, providing guarantees on optimization dynamics.

● **Benchmarking the method.** Conducting experiments on 22 benchmark PDEs using vanilla *PINN*, we demonstrate that our approach improves the quality compared to existing optimizers. The proposed algorithm achieves SOTA in **77.3%** of cases, against $27.3\%$ for the second best.

● **Extensive empirical study.** We show that the proposed weighting scheme reduces the conflict of gradient magnitudes better than its competitors. We attribute this as the primary reason for dominance of our approach across the majority of PDEs. Additionally, we analyze the computational overhead and examine the robustness of our algorithm to changes in hyperparameters.

● **Challenge-test.** We demonstrate the scalability of the proposed method in real-world scenarios. Specifically, we train the *DoMINO* model (Ranade et al., 2025) to solve the three-dimensional Navier–Stokes equation on a sample vehicle from the *DrivAerML* dataset (Ashton et al., 2024).

## 4 SETUP

### 4.1 ASSUMPTIONS

Since our study is motivated by the real-world problem, we address the most general case possible. First, we require the objective to be smooth with respect to the Euclidean norm.

**Assumption 1.** *The function $\mathcal{L}(\theta, \pi)$ is L-smooth, i.e. for all $(\theta_1, \pi_1), (\theta_2, \pi_2) \in \mathbb{R}^d \times S$ it satisfies*

$$\|\nabla \mathcal{L}(\theta_1, \pi_1) - \nabla \mathcal{L}(\theta_2, \pi_2)\| \leq L\|(\theta_1, \pi_1) - (\theta_2, \pi_2)\|.$$

Lipschitz continuity of the gradient is commonly imposed in prior work on *PINN*s (Li et al., 2023, Assumption 1), (Hwang and Lim, 2024, Theorem 4.5), (Wu et al., 2024, Assumption 3.2), (Liu et al., 2024a, Theorem 1). While this assumption is generally unrealistic for neural networks (Cybenko, 1989), it is necessary to construct any theory for Deep Learning (Patel and Berahas, 2024).

To enable more accurate selection of the weights $\pi$, we account for the geometry of $S$ by utilizing the Bregman divergence (Nemirovskij and Yudin, 1983).

**Definition 1.** *The Bregman divergence corresponding to the distance generating function $\psi : S \to \mathbb{R}$ is defined as*

$$D_\psi(\pi_1, \pi_2) = \psi(\pi_1) - \psi(\pi_2) - \langle \nabla \psi(\pi_2), \pi_1 - \pi_2 \rangle.$$

Earlier, we mentioned the example where $D_\psi$ is the Kullback-Leibler divergence. This is particularly significant for the purposes of this paper, as we choose $S$ as the unit simplex. However, the theory is established in the general case. Analysis of the problem (2) requires $D_\psi$ to have several basic properties. In particular, Definition 2 is valid only if $D_\psi$ is bounded from below on $S$. In the following, we present an assumption regarding the distance generating function.

**Assumption 2.** *The function $\psi$ is **1-strongly convex**, i.e. for all $\pi_1, \pi_2 \in S$ it satisfies*

$$\psi(\pi_1) \geq \psi(\pi_2) + \langle \nabla \psi(\pi_2), \pi_1 - \pi_2 \rangle + \frac{1}{2}\|\pi_2 - \pi_1\|^2.$$

Note that this assumption does not reduce the class of neural networks under consideration, as it is solely related to the choice of regularizer. Additionally, it holds for all commonly used divergences.

### 4.2 PROPERTIES OF THE OBJECTIVE

In this section, we obtain several properties of the objective by leveraging its structure.

**Lemma 1.** *Consider the problem (2) under Assumption 2. Then, for every $\theta \in \mathbb{R}^d$ the function $\mathcal{L}(\theta, \pi)$ is $\lambda$-**strongly concave**, i.e. for all $\pi_1, \pi_2 \in S$ it satisfies*

$$\mathcal{L}(\theta, \pi_1) \leq \mathcal{L}(\theta, \pi_2) + \langle \nabla_\psi \mathcal{L}(\theta, \pi_2), \pi_1 - \pi_2 \rangle$$

$$- \frac{\lambda}{2}\left(D_\psi(\pi_1, \pi_2) + D_\psi(\pi_2, \pi_1)\right).$$

See the proof in Appendix E. Lemma 1 shows that the problem (2) is a nonconvex-strongly concave SPP. Hence, it has a single maximum $\pi^*(\theta)$ on $S$ for every fixed value of $\theta$.

### 4.3 OPTIMALITY CONDITION

It is challenging to analyze N-SC SPPs using the usual definition of a stationary point. Instead, prior works equivalently reduce it to a stationary point of a minimization problem (Huang et al., 2021):

$$\Phi(\theta) = \mathcal{L}(\theta, \pi^*(\theta)).$$

Since $S$ is a bounded convex set, Danskin's theorem implies that $\Phi$ is differentiable with $\nabla\Phi(\theta) = \nabla_\theta\mathcal{L}(\theta, \pi^*(\theta))$ (Rockafellar, 2015). The common convergence metric employed in the literature is the following (Zhang et al., 2021; Wang et al., 2024; Xu et al., 2024).

**Definition 2.** *($\varepsilon$-stationary point) of $\Phi(\theta)$. A point $\theta$ is an $\varepsilon$-stationary point of $\Phi$, if*
$$\|\nabla\Phi(\theta)\| \leq \varepsilon.$$

For N-SC SPPs, convergence in the sense of Definition 2 implies convergence to a stationary point in the standard sense used for SPPs (Lin et al., 2020, Proposition 4.12).

## 5 ALGORITHMS AND ANALYSIS

### 5.1 MAIN ALGORITHM

In this section, we follow the trend of investigating N-SC SPPs through modifications of `TT-GDA`. Adapting it to the problem (2), we present **B**regman **G**radient **D**escent **A**scent.

Due to the complex landscape of the problem to be solved, the algorithmic schemes we rely on are extremely simple. Since the parameters $\theta$ may take any value, it suffices to use the classic gradient descent step (Nemirovskij and Yudin, 1983) to update them (Line 4). However, the weights are selected from a convex bounded set described by Non-Euclidean geometry. Consequently, we utilize the Bregman proximal mapping (Nemirovskij and Yudin, 1983) to perform the ascent step (Line 5). The subproblem in

---

**Algorithm 1** BGDA

1: **Input:** Starting point $(\theta^0, \pi^0) \in \mathbb{R}^d \times S$, number of iterations $T$
2: **Parameters:** Stepsizes $\gamma_\theta, \gamma_\pi > 0$
3: **for** $t = 0, \dots, T-1$ **do**
4: $\quad \theta^{t+1} = \theta^t - \gamma_\theta \nabla_\theta \mathcal{L}(\theta^t, \pi^t)$
5: $\quad \pi^{t+1} = \arg\min_{\pi \in S} \{q(\pi)\}$, where
$$q(\pi) = -\gamma_\pi \left\langle \nabla_\pi \mathcal{L}(\theta^t, \pi^t), \pi \right\rangle + D_\psi(\pi, \pi^t)$$
6: **end for**

---

Line 5 requires estimating statistics of the objective only once and therefore does not pose any significant computational difficulties compared to the basic descent step. Moreover, it often has a closed-form solution. For example, if $D_\psi$ is the KL-divergence (Nemirovskij and Yudin, 1983), then

$$\pi^{t+1} = \left( \frac{\exp\{\gamma_\pi (\nabla_\pi \mathcal{L}(\theta^t, \pi^r))_i\}}{\sum_{i=1}^M \exp\{\gamma_\pi (\nabla_\pi \mathcal{L}(\theta^t, \pi^r))_i\}} \right)_{i=1}^M.$$

In the analysis of Algorithm 1, it is fundamental to utilize steps of varying sizes. One possible explanation is that the landscape of the objective is much better in the strongly concave component. Consequently, more confident steps can be taken to update the weights. The primary theoretical challenge in the analysis of the method is to show the convergence of the iterative scheme based on the metric given in Definition 2. Indeed, for each value of the model parameters $\theta^t$ there is an optimal point $\pi^*(\theta^t)$. To address the technical difficulties, we must show that the method generates a sequence of points $\{(\theta^t, \pi^t)\}_{t=1}^T$ for which the distance between $\pi^t$ and $\pi^*(\theta^t)$ decreases when increasing $t$. Moreover, we have to account for the non-Euclidean geometry of the problem.

**Lemma 2.** *Consider the problem 2 under Assumptions 1, 2. Then, Algorithm 1 produces such $\{(\theta^t, \pi^t)\}_{t=1}^T$, that*

$$D_\psi(\pi^*(\theta^{t+1}), \pi^{t+1}) \leq \left(1 - \frac{1}{64\kappa^2}\right) D_\psi(\pi^*(\theta^t), \pi^t) + 264\gamma_\theta^2 \kappa^6 \|\nabla\Phi(\theta^t)\|^2,$$

*where $\kappa = L/\lambda$ is the condition number of $\mathcal{L}(\theta, \pi)$ in $\pi$.*

See the proof in Appendix F. Lemma 2 shows how the distance between the current weight iterate $\pi^t$ and the ideal response $\pi^*(\theta^t)$ evolves over time. This is a key result needed to prove convergence. Indeed, since we consider the Euclidean setting in the nonconvex variables $\theta$, the standard inexact gradient descent analysis implies

$$\Phi(\theta^{t+1}) - \Phi(\theta^0) \leq -\Omega(\gamma_\theta) \left( \sum_{t=1}^{T-1} \|\nabla\Phi(\theta^t)\|^2 \right) + \mathcal{O}\left(\gamma_\theta L^2\right) \sum_{t=1}^{T-1} D_\psi(\pi^*(\theta^t), \pi^t).$$

Thus, for a sufficiently small step $\gamma_\theta$, it is guaranteed to neglect the inaccuracy of finding the maximum at the ascent step. By carefully evaluating $D_\psi(\pi^*(\theta^t), \pi^t)$ from above and selecting appropriate $\gamma_\theta$, the convergence is obtained. We formulate this fact as a main theorem.

**Theorem 1.** *Consider the problem 2 under Assumptions 1, 2. Then, Algorithm 1 requires*

$$\mathcal{O}\left(\frac{\kappa^4 L \Delta + \kappa^2 L^2 D_\psi(\pi^*(\theta^0), \pi^0)}{\varepsilon^2}\right) \text{ iterations}$$

*to achieve an arbitrary $\varepsilon$-solution, where $\varepsilon^2 = \frac{1}{T}\sum_{t=1}^{T-1}\|\nabla\Phi(\theta^t)\|^2$, $\Delta = \Phi(\theta^0) - \Phi(\theta^*)$. $\kappa = L/\lambda$ is the condition number of $\mathcal{L}(\theta, \pi)$ in $\pi$.*

See the proof in Appendix G. Note that the derived estimate of $T$ is worse than that obtained in (Huang et al., 2021) for the Euclidean setting. However, if $S$ is a unit simplex intersected with a euclidean ball, it can be significantly improved $\mathcal{O}\left(\kappa L/\varepsilon^2\right)$ (see Appendix H for the detailed discussion).

## 5.2 Practical Version of BGDA

Since neural networks exhibit a complex loss landscape, it is common practice to run adaptive versions of algorithms, even when their theoretical guarantees do not account for such modifications. Following this trend, we develop an adaptive modification of Algorithm 1. In Algorithm 2, the gradient $\nabla_\theta \mathcal{L}(\theta^t, \pi^t)$ is smoothed with its previous values as a running average (Line 4). In practice, this approach aids in identifying a suitable descent direction more quickly. Furthermore, we propose accumulating the gradient history to vary the step sizes (Lines 5, 6). This method is effective, as the gradient magnitude indicates the loss smoothness locally, which leads to more confident steps and faster convergence. A practice-driven bias correction of calculated statistics is also implemented (Lines 7, 8, 9). To update model parameters and weights, Algorithm 2 performs the descent-ascent scheme, identical to Algo-

---

**Algorithm 2** `AdaptiveBGDA` (`AdaBGDA`)

1: **Input:** Starting point $(\theta^0, \pi^0) \in \mathbb{R}^d \times S$, number of iterations $T$
2: **Parameters:** Stepsizes $\gamma_\theta, \gamma_\pi > 0$
3: **for** $t = 0, \ldots, T-1$ **do**
4: $\quad m_\theta^{t+1} = \alpha_1 m_\theta^t + (1 - \alpha_1)\nabla_\theta \mathcal{L}(\theta^t, \pi^t)$
5: $\quad v_\theta^{t+1} = \alpha_2 v_\theta^t + (1 - \alpha_2)(\nabla_\theta \mathcal{L}(\theta^t, \pi^t))^2$
6: $\quad v_\pi^{t+1} = \beta v_\pi^t + (1 - \beta)(\nabla_\pi \mathcal{L}(\theta^t, \pi^t))^2$
7: $\quad \widehat{m}_\theta^{t+1} = \frac{m_\theta^{t+1}}{1 - \alpha_1^t}$
8: $\quad \widehat{v}_\theta^{t+1} = \frac{v_\theta^{t+1}}{1 - \alpha_2^t}$
9: $\quad \widehat{v}_\pi^{t+1} = \frac{v_\pi^{t+1}}{1 - \beta^t}$
10: $\quad \theta^{t+1} = \theta^t - \gamma_\theta \frac{\widehat{m}_\theta^{t+1}}{\widehat{v}_\theta^{t+1}}$
11: $\quad \pi^{t+1} = \arg\min_{\pi \in S}\{q(\pi)\}$, where
$\quad q(\pi) = -\gamma_\pi\left\langle \widehat{m}_\pi^{t+1}/\widehat{v}_\pi^{t+1}, \pi \right\rangle + D_\psi(\pi, \pi^t)$
12: **end for**

---

rithm 1. Namely, `AdaptiveBGDA` utilizes `Adam` (Kingma and Ba, 2014) and `RMSProp` (Xu et al., 2021) to perform descent and ascent steps, respectively. See Table 2 for justification.

## 6 Numerical Experiments

In Sections 6.1-6.4, we employ a vanilla *PINN* with 5 hidden layers of size 100. We initialize hyperparameters of Algorithm 2 as $\gamma_\pi^0 = 0.1$, $\gamma_\theta^0 = 0.008$, $\alpha_1^0 = 0.9$, $\alpha_2^0 = 0.999$, $\beta^0 = 0.999$, $\lambda = 0.01$. $\gamma_\theta$ is scheduled linearly with the final value of $0.0004$. To tune the baselines from *PINNacle* (Hao et al., 2023), we follow the authors' recommendations; for the remaining methods, hyperparameters are selected using the same procedure as for our algorithm. Empirical analysis is conducted with NVIDIA TESLA A100 with 80 GB of GPU memory.

In Section 6.5, we work with the *DoMINO* model developed by NVIDIA (Ranade et al., 2025). To train `Adam`, we use tuning provided in the code attached to (Contributors, 2023). In experiments with Algorithm 2, we initialize hyperparameters as $\gamma_\pi^0 = 0.1$, $\gamma_\theta^0 = 0.002$, $\alpha_1^0 = 0.9$, $\alpha_2^0 = 0.999$, $\beta^0 = 0.999$, $\lambda = 0.1$. $\gamma_\theta$ decreases linearly to $0,001$ during $500$ epochs. For this part of empirical study, we use NVIDIA H200 with 140 GB of GPU memory.

## 6.1 Exploring Variants of Adaptivity

During the empirical study, we used *Poisson 2d-C* to test various combinations of adaptive techniques, such as `Adam` (Kingma and Ba, 2014) and `RMSProp` (Xu et al., 2021). It was `Adam`+`RMSProp` that turned out to be the best one. We attribute this to

Table 2: Comparison of approaches to adaptivity in Algorithm 1. Mean **L2RE** over 3 runs is used as a quality metric.

| **Approach** | Adam+RMS | Adam+Adam | RMS+RMS |
|---|---|---|---|
| **L2RE** | **8.15E-3** | 4.45E-2 | 6.02E-1 |

the fact that `Adam` allows to account for the poor loss landscape in $\theta$ via gradient smoothing, while the landscape in $\pi$ is strongly convex, and steps along the current gradient are more appropriate.

## 6.2 VALIDATION ON *PINNacle* BENCHMARK

We provide an extensive comparison of `AdaBGDA` (Algorithm 2) with existing approaches. To evaluate the learning potential and generalization capabilities of our approach, we consider 22 partial differential equations sourced from *PINNacle* (Hao et al., 2023) that covers a broad spectrum of PDEs: complex geometry, multiple domains, varying coefficients, long time. As competitors, we consider all methods presented in *PINNacle* (Hao et al., 2023): `LBFGS` (Byrd et al., 1995), `Adam` (Kingma and Ba, 2014), `MultiAdam` (Yao et al., 2023), and combinations of `Adam` with `RAR` (Lu et al., 2021), `LRA` (Wang et al., 2021), `NTK` (Wang et al., 2022).

Table 3: Comparison of `AdaBGDA` (ours) to the existing techniques. In all experiments, the model is trained to the performance limit. **L2RE** is used as a quality metric. We highlight the best and the second best results for each PDE.

| PDE | | Optimizer | | | | | | |
|---|---|---|---|---|---|---|---|---|
| | | Adam | LBFGS | LRA | NTK | RAR | MultiAdam | AdaBGDA |
| *Burgers* | 1d-C | (1.44±0.04)E-2 | (1.33±0.01)E-2 | (2.66±0.33)E-2 | (1.90±0.02)E-2 | (3.10±0.32)E-2 | (4.96±0.38)E-2 | **(1.29±0.01)E-2** |
| | 2d-C | (2.72±0.32)E-1 | (4.68±0.08)E-1 | **(2.58±0.13)E-1** | (2.83±0.31)E-1 | (3.42±0.24)E-1 | (3.26±0.46)E-1 | (4.20±0.10)E-1 |
| *Poisson* | 2d-C | (3.41±0.15)E-2 | NaN | (1.11±0.09)E-1 | (1.14±0.11)E-2 | (7.53±0.62)E-1 | (2.79±0.25)E-2 | **(8.15±0.20)E-3** |
| | 2d-CG | (5.50±0.61)E-2 | (2.93±0.04)E-1 | (4.11±0.24)E-2 | (1.35±0.12)E-2 | (6.64±0.50)E-1 | (2.76±0.19)E-1 | (1.70±0.51)E-2 |
| | 3d-CG | (3.94±0.21)E-1 | (7.20±0.16)E-1 | (1.08±0.07)E-1 | (8.73±1.32)E-1 | (5.55±0.38)E-1 | (3.56±0.43)E-1 | **(6.41±0.21)E-2** |
| | 2d-MS | (6.64±0.49)E-1 | (1.46±0.01)E+0 | (7.84±0.65)E-1 | (7.90±0.44)E-1 | (6.52±0.35)E-1 | (6.23±0.33)E-1 | **(3.43±0.08)E-1** |
| *Heat* | 2d-VC | (2.58±0.27)E-1 | (2.28±0.14)E-1 | (2.13±0.29)E-1 | **(2.07±0.21)E-1** | (1.05±0.10)E+0 | (4.94±0.56)E-1 | (2.99±0.19)E-1 |
| | 2d-MS | (6.71±0.60)E-2 | (1.74±0.10)E-2 | (8.65±1.21)E-2 | (4.31±0.46)E-2 | (7.93±0.53)E-2 | (2.05±0.18)E-1 | **(1.40±0.35)E-2** |
| | 2d-CG | (3.83±0.47)E-2 | (8.54±0.17)E-1 | (1.16±0.12)E-1 | (1.20±0.10)E-1 | (2.58±0.17)E-2 | (7.68±0.69)E-2 | **(2.49±0.11)E-2** |
| | 2d-LT | (9.98±0.01)E-1 | (1.00±0.00)E+0 | (9.97±0.02)E-1 | (1.00±0.00)E+0 | (9.98±0.04)E-1 | (9.98±0.04)E-1 | **(9.96±0.01)E-1** |
| *NS* | 2d-C | (4.67±0.35)E-2 | (2.11±0.05)E-1 | NaN | (2.01±0.23)E-1 | (4.51±0.31)E-1 | (7.03±0.75)E-1 | **(2.35±0.59)E-2** |
| | 2d-CG | (1.18±0.12)E-1 | NaN | (3.22±0.32)E-1 | (2.66±0.30)E-1 | (3.26±0.21)E-1 | (4.51±0.33)E-1 | **(7.12±0.27)E-2** |
| | 2d-LT | (9.91±0.41)E-1 | **(9.70±0.07)E-1** | (9.90±0.05)E-1 | (9.99±0.01)E-1 | (9.99±0.01)E-1 | (1.00±0.00)E+0 | (9.70±0.08)E-1 |
| *Wave* | 1d-C | (2.83±0.18)E-1 | NaN | (3.65±0.36)E-1 | (9.20±0.82)E-2 | (5.62±0.57)E-1 | (1.21±0.10)E-1 | **(1.63±0.46)E-2** |
| | 2d-CG | (1.66±0.02)E+0 | (1.33±0.00)E+0 | (1.53±0.10)E+0 | (2.09±0.15)E+0 | (1.21±0.09)E+0 | (1.08±0.02)E+0 | **(7.80±0.03)E-1** |
| | 2d-MS | (1.02±0.01)E+0 | (1.36±0.01)E+0 | (9.97±0.36)E-1 | (1.03±0.04)E+0 | (1.32±0.08)E+0 | (1.01±0.01)E+0 | **(9.35±0.08)E-1** |
| *Chaotic* | GS | (1.58±0.00)E-1 | NaN | (9.76±0.05)E-2 | (2.16±0.00)E-1 | (9.10±0.74)E-2 | (9.36±0.00)E-2 | **(9.29±0.00)E-2** |
| | KS | (9.94±0.09)E-1 | NaN | (9.58±0.03)E-1 | (9.61±0.05)E-1 | (1.02±0.01)E+0 | (9.69±0.10)E-1 | **(9.51±0.02)E-1** |
| *High dim* | PNd | (2.66±0.09)E-3 | (4.69±0.13)E-4 | (4.87±0.58)E-4 | (4.77±0.20)E-3 | (3.37±0.26)E-3 | (4.08±0.11)E-3 | **(1.31±0.16)E-4** |
| | HNd | (3.67±0.00)E-1 | **(1.13±0.10)E-4** | (3.92±0.07)E-1 | (3.98±0.01)E-1 | (3.71±0.21)E-1 | (3.00±0.04)E-1 | (1.35±0.15)E-4 |
| *Inverse* | PInv | (1.03±0.13)E-1 | NaN | (1.66±0.15)E-1 | (1.77±0.23)E-1 | (9.53±0.57)E-2 | (1.32±0.08)E-1 | **(6.11±0.22)E-2** |
| | HInv | (5.23±0.29)E-2 | NaN | (5.08±0.07)E-2 | (7.77±0.38)E-2 | (1.59±0.11)E+0 | (7.87±0.35)E-2 | **(4.33±0.27)E-2** |

Table 3 shows that `AdaBGDA` is dominant in **77.3%** of cases. In **18%** of its rows, we outperform the closest competitor by more than a factor of two; in **36%** of its rows, `AdaBGDA` improves **L2RE** by more than **30%**. At the same time, the previous best optimizer dominated only 27.3% of the benchmark (**2.83 times worse** than our result), and among those cases, only one out of 22 PDEs shows an improvement larger than **5.4%**.

For certain PDEs (*Heat 2d-LT*, *NS 2d-LT*, *Wave 2d-MS*), `AdaBGDA` yields large **L2RE** values, similarly to other optimizers. We attribute this behavior to the fact that the vanilla *PINN* architecture is too simple for such problems. To demonstrate that the saddle-point formulation of *PINN* training has the potential to consistently improve model quality across a wide range of PDEs even in the case of more expressive architectures, we train a state-of-the-art model to solve a three-dimensional Navier–Stokes problem in a domain with complex geometry in Section 6.5.

## 6.3 EXPLORING THE CONFLICTING GRADIENTS

Table 3 illustrates the stability of the proposed method under changes in problem type, boundary/initial conditions, and domain geometry. To numerically investigate this phenomenon, we measure the ratio $\chi = \|\nabla \mathcal{L}_r(\theta)\| / \|\nabla \mathcal{L}_b(\theta)\|$ while solving *Poisson 2d-C*. We break the iterations into groups $I_1 = [0, 10000)$, $I_2 = [10000, 20000)$, $I_3 = [20000, 30000]$ and examine the distributions of $\chi_1$, $\chi_2$, $\chi_3$, including their means $\overline{\chi}_1$, $\overline{\chi}_2$, $\overline{\chi}_3$ and variances $\sigma_1$, $\sigma_2$, $\sigma_3$.

In Figure 1, one can see the dynamics of `NTK` (Wang et al., 2021). This optimizer is state-of-the-art for the selected PDE. From the first epochs, $\|\nabla\mathcal{L}_r(\theta)\|$ demonstrates significant superiority over $\|\nabla\mathcal{L}_b(\theta)\|$. At this stage, we observe $\overline{\chi}_1 = 2487$, $\sigma_1 = 2352$. During the next group of iterations, these ratios hold approximately at the same level $\overline{\chi}_2 = 2342$, $\sigma_2 = 1628$; and after another 10000 they decrease to $\overline{\chi}_3 = 1998$, $\sigma_3 = 1360$. Thus, at the beginning of optimization, the value of $\chi$ rapidly concentrates extremely far away from the desired case of equal magnitudes and then slowly decreases. Consequently, *PINN* overfits to the boundary condition.

The training process of our method is significantly more stable. Figure 2 shows results for the proposed `AdaptiveBGDA`. Using this scheme, we obtain $\overline{\chi}_1 = 7$, $\sigma_1 = 7$; $\overline{\chi}_2 = 25$, $\sigma_2 = 27$; $\overline{\chi}_3 = 45$, $\sigma_3 = 127$. The pathology is much less pronounced. The resulting improvement is statistically significant. Indeed, for $I_1$ only $\approx 9\%$ of the values obtained with `NTK` fall within the $3\sigma_1$-interval for `AdaptiveBGDA`. At the same time, for $I_2$ and $I_3$ such values do not exist at all.

The superiority of our method is particularly well demonstrated by the error heat maps. Such a comparison is presented in Figure 3. In the right part of Figure 3, we observe a significant region within the interior of the domain where the approximated solution exhibits a large error. The absence of such a region on the left side of Figure 3 illustrates that we successfully address the issue of underestimating losses in the interior of the domain.

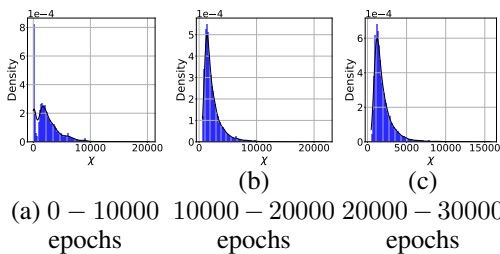

(a) $0 - 10000$ epochs    (b) $10000 - 20000$ epochs    (c) $20000 - 30000$ epochs

Figure 1: Dynamics of $\chi = \|\nabla\mathcal{L}_r(\theta)\|/\|\nabla\mathcal{L}_b(\theta)\|$ during optimization via `NTK`. The experiment is made on *Poisson 2d-C*.

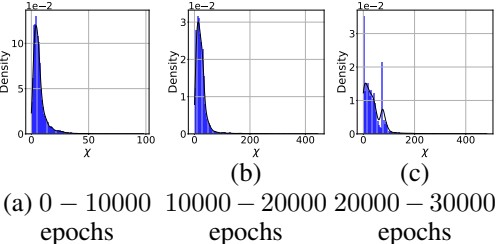

(a) $0 - 10000$ epochs    (b) $10000 - 20000$ epochs    (c) $20000 - 30000$ epochs

Figure 2: Dynamics of $\chi = \|\nabla\mathcal{L}_r(\theta)\|/\|\nabla\mathcal{L}_b(\theta)\|$ during optimization via `AdaBGDA` (ours). The experiment is made on *Poisson 2d-C*.

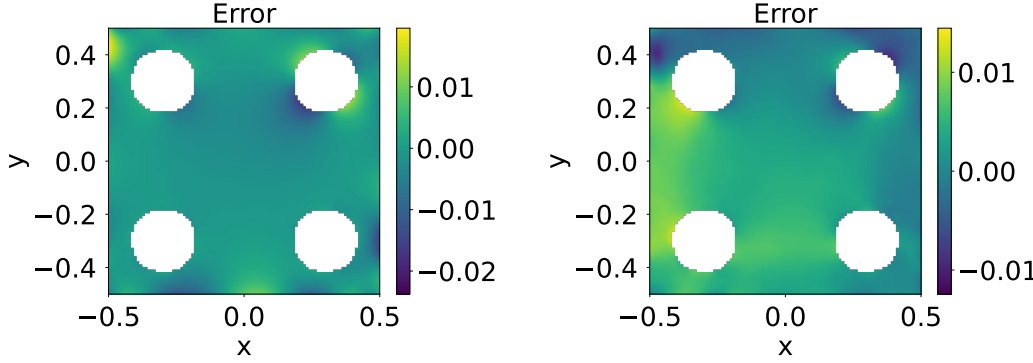

Figure 3: Heat maps of signed relative errors of *PINN* trained to solve *Poisson 2d-C*. `AdaptiveBGDA` (left) is compared with `NTK` (right).

## 6.4 EXPLORING THE COMPUTATIONAL OVERHEAD

One of the key characteristics of an optimizer is the trade-off between performance and computational overhead. Since `AdaBGDA` (Algorithm 2) includes an additional update to weight competing losses, conducting such a study is particularly important.

Figure 4 shows a direct comparison of the actual runtime of `AdaBGDA` and its competitors on the *Wave 1d-C* problem. `AdaBGDA` achieves convergence approximately **2.5** times faster than state-of-the-art scheme for this PDE. The intersection of deviations at the beginning of training is associated with the rapid convergence of methods. Notably, the model reaches a higher final performance when trained with the proposed algorithm.

Table 4: Comparison of time/space complexity of AdaBGDA and competing methods on *Burgers 1d-C*: (a) – Adam, (b) – LBFGS, (c) – LRA, (d) – NTK, (e) – RAR, (f) – MultiAdam, (g) – SSBroyden, (h) – NNCG, (i) – AdaBGDA. The second row of the table shows the time (sec) for 1000 iterations. The third row shows the peak GPU utilization (GB) on storing the optimizer states.

| Method | (a) | (b) | (c) | (d) | (e) | (f) | (g) | (h) | (i) |
|---|---|---|---|---|---|---|---|---|---|
| Time | 8.24 | 562.07 | 22.43 | 19.90 | 8.33 | 14.09 | 176.69 | 13937.6 | 7.64 |
| Space | 0.23 | 0.26 | 0.50 | 0.47 | 0.25 | 0.45 | 10.66 | 2.68 | 0.37 |

We also provide an ablation study of time-per-iteration and memory consumption (see Table 4). We restrict this experiment to a single run, since all runs were performed on a single GPU, and the contribution of noise sources, such as variations in temperature, is negligible. Additionally, we use several methods that are not the part of *PINNacle*, namely SSBroyden (Kiyani et al., 2025) and NNCG (Rathore et al., 2024). Below, we formulate the list of core observations and insights.

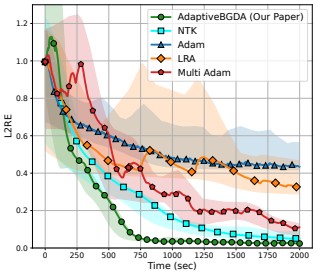

• AdaBGDA does not experience an increase in iteration time. Indeed, in the case of the unit simplex with KL-divergence, the ascent Bregman step has a closed-form expression in terms of the values of the objective components. Thus, updating the weights requires only a forward pass, which is already performed for updating the model parameters. Consequently, AdaBGDA does not incur higher computational cost than first-order methods such as Adam.

• GPU utilization also does not increase compared to competing methods. We attribute this to the fact that the number of model parameters ($40K$ in our experiments) is significantly larger than the number of weights (no more than 11 in *PINNacle*). Consequently, optimizer states for the weights do not inflate memory requirements.

Figure 4: Comparison of AdaptiveBGDA to competitors on *Wave 1d-C*.

In light of the above, we suggest that saddle-point based approaches have potential to be as computationally efficient as Adam while achieving quality comparable to LRA/NTK.

## 6.5 EXPLORING THE SCALABILITY

Previous sections demonstrate superiority of AdaBGDA over competing methods in terms of **L2RE**, training fairness, and computational/memory efficiency. However, these results are obtained with vanilla *PINN*s, which are hardly usable in real-world scenarios. Consequently, an important question remains open: *is the proposed approach scalable to complex modern architectures?*

To address this issue, we consider an automotive aerodynamics as a challenge-test problem. We fix a single vehicle geometry from the *DrivAerML* dataset (Ashton et al., 2024) and solve the three-dimensional Navier–Stokes equation for incompressible fluid to predict the velocity field and volume/surface pressure. Fluid density and kinematic viscosity are $\rho = 1.226$ and $\nu = 1.507 \cdot 10^{-5}$, respectively. We use the *DoMINO* model (Ranade et al., 2025), a $38M$ point-cloud-based architecture that leverages local geometric information to predict flow fields at discrete points.

Table 5: Comparison of AdaBGDA (ours) and competitors on *3d Navier-Stokes*. We measure **L2RE** of: (a) – x-velocity, (b) – y-velocity, (c) – z-velocity, (d) – volume pressure, (e) – surface pressure.

| Metric | (a) | (b) | (c) | (d) | (e) |
|---|---|---|---|---|---|
| Adam | (3.39±0.10)E-1 | (8.60±0.73)E-1 | (7.16±0.23)E-1 | (4.55±0.23)E-1 | (2.71±0.05)E-1 |
| LBFGS | (3.62±0.23)E-1 | (9.56±0.51)E-1 | (8.23±0.51)E-1 | (4.88±0.28)E-1 | (3.42±0.41)E-1 |
| AdaBGDA | **(2.78±0.14)E-1** | **(5.99±0.35)E-1** | **(5.34±0.13)E-1** | **(2.89±0.27)E-1** | **(2.69±0.02)E-1** |

Table 5 shows that employing the SPP approach leads to a fair and robust training process not only in simple settings, but also for large-scale models and complex PDEs. While preserving the quality of surface pressure predictions, AdaBGDA achieves statistically significant improvements

across the remaining metrics compared to competing methods. Thus, the effect of balance across all components of the objective is preserved as model scale and task complexity increase.

## 6.6 OTHER EXPERIMENTS

Beyond the main empirical study, we also present several additional results. Appendix A reports experiments with several variants of the vanilla *PINN* evaluated on PDEs from *PINNacle*. In Appendix B, `AdaBGDA` is compared to competing saddle-point approaches: `dual-dimer` (Liu and Wang, 2021) and `AL-PINN` (Son et al., 2023). Our scheme dominates in 77.3% and 63.6% of the cases, respectively. Appendix C examines the robustness of `AdaBGDA` to hyperparameter tuning. Finally, Appendix D provides a comparison between theoretical and empirical convergence of `AdaBGDA`, demonstrating that numerical observations are consistent with theoretical guarantees.

## 7 CONCLUSION

In this paper, we observe that existing weighting schemes for *PINN* training do not achieve a fully balanced optimization process. To address this issue, we reformulate the training problem as the nonconvex-strongly concave SPP of non-Euclidean nature. In addition to theoretical analysis, we conduct a comprehensive empirical study. We observe a significant increase in the model quality (Table 3) while preserving the computational efficiency of optimizer (Table 4). We empirically evaluate an increase in the fairness of the training process (Figures 1, 2). Experiments also demonstrate the scalability of the proposed scheme (Table 5). Thus, the ability of the saddle-point approach to balance conflicting loss terms is crucial for reducing the cost of optimizer selection. As a result, further research in this direction holds promise for developing robust training frameworks for *PINN*s, improving their applicability to real-world problems.

## ACKNOWLEDGMENTS

The work was supported by the Ministry of Economic Development of the Russian Federation (agreement No. 139-15-2025-013, dated June 20, 2025, IGK 000000C313925P4B0002).

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

APPENDIX

CONTENTS

To ensure reproducibility, we attach the code: `https://anonymous.4open.science/r/pinns-bgda-00D6`

## A  ADDITIONAL EXPERIMENTS

In this section, we provide additional information to accompany the work. In addition, we use some variations of vanilla *PINN* to validate the theoretical insights.

• *gPINN*. It is known that the residual $(\mathcal{R}_i[u]-f_i)(x)$ must be zero inside the domain. Consequently, its derivative must also be equal to zero. This approach proposes to modify the objective by adding $\|\partial/\partial x (\mathcal{R}_i[u]-f_i)(x)\|^2$ as a regularization. In (Yu et al., 2022), it is shown that *gPINN* has improved quality of the approximation inside the domain $\Omega$.

• *GAAF*. This architecture relies on adaptive activation functions (both layer- and neuron-wise). (Jagtap et al., 2020b) demonstrates the advantages of this approach over vanilla *PINN*s.

• *LAAF*. Considers *GAAF* with slope recovery term. For the details, see (Jagtap et al., 2020a).

Table 6: Comparison of `AdaBGDA` (ours) to the existing techniques. In all experiments, the model is trained to the performance limit. **L2RE** is used as a quality metric. We highlight the **best** results for each pair.

| PDE | | gPINN | | LAAF | | GAAF | |
|---|---|---|---|---|---|---|---|
| | | Best | Ours | Best | Ours | Best | Ours |
| Burgers | 1d-C | (2.21±0.03)E-1 | **(1.34±0.02)E-2** | (1.46±0.02)E-2 | **(1.28±0.02)E-2** | (5.15±0.06)E-2 | **(1.31±0.02)E-2** |
| | 2d-C | **(3.32±0.22)E-1** | (5.05±0.09)E-1 | **(2.74±0.05)E-1** | (4.38±0.07)E-1 | **(2.97±0.06)E-1** | (5.02±0.08)E-1 |
| Poisson | 2d-C | (6.83±0.40)E-1 | **(5.69±0.35)E-1** | (7.61±0.49)E-1 | **(1.45±0.11)E-2** | (6.10±0.44)E-1 | **(4.52±0.31)E-3** |
| | 2d-CG | (8.00±0.60)E-1 | **(4.37±0.28)E-1** | (4.73±0.24)E-1 | **(1.16±0.09)E-2** | (8.61±0.66)E-1 | **(2.76±0.22)E-2** |
| | 3d-CG | **(4.78±0.31)E-1** | (5.70±0.35)E-1 | (5.67±0.32)E-1 | **(5.64±0.41)E-2** | (5.12±0.28)E-1 | **(9.08±0.57)E-2** |
| | 2d-MS | (6.17±0.33)E-1 | **(4.61±0.23)E-1** | (6.03±0.32)E-1 | **(3.74±0.27)E-1** | (9.17±0.21)E-1 | **(4.14±0.28)E-1** |
| Heat | 2d-VC | (2.12±0.13)E+0 | **(1.00±0.08)E+0** | (6.57±0.42)E-1 | **(2.55±0.18)E-1** | (8.60±0.51)E-1 | **(7.07±0.43)E-1** |
| | 2d-MS | (1.13±0.07)E-1 | **(3.94±0.32)E-2** | (7.54±0.66)E-2 | **(1.92±0.12)E-2** | (9.83±0.71)E-1 | **(6.67±0.55)E-2** |
| | 2d-CG | **(9.47±0.75)E-2** | (1.08±0.08)E-1 | **(2.43±0.20)E-2** | (4.03±0.34)E-2 | (4.64±0.31)E-1 | **(1.23±0.16)E-2** |
| | 2d-LT | (1.00±0.00)E+0 | **(9.98±0.01)E-1** | (9.99±0.01)E-1 | **(9.97±0.01)E-1** | (9.99±0.01)E-1 | **(9.97±0.01)E-1** |
| NS | 2d-C | (7.82±0.62)E-2 | **(6.14±0.55)E-2** | **(3.72±0.34)E-2** | (8.21±0.62)E-2 | (3.82±0.33)E-2 | **(2.63±0.24)E-2** |
| | 2d-CG | (1.52±0.15)E-1 | **(1.11±0.08)E-1** | **(8.52±0.64)E-2** | (1.26±0.09)E-1 | (1.70±0.12)E-1 | **(1.05±0.08)E-1** |
| | 2d-LT | (9.94±0.02)E-1 | **(9.62±0.03)E-1** | **(9.98±0.01)E-1** | (9.99±0.01)E-1 | (9.99±0.01)E-1 | (9.99±0.01)E-1 |
| Wave | 1d-C | (5.40±0.41)E-1 | **(7.22±0.12)E-1** | (4.62±0.31)E-1 | **(2.61±0.23)E-1** | (6.84±0.52)E-1 | **(3.03±0.24)E-2** |
| | 2d-CG | (8.11±0.06)E-1 | **(7.81±0.04)E-1** | (8.23±0.06)E-1 | **(7.91±0.05)E-1** | (7.91±0.05)E-1 | **(7.83±0.03)E-1** |
| | 2d-MS | (1.00±0.00)E+0 | **(9.11±0.06)E-1** | (1.04±0.08)E+0 | **(9.93±0.07)E-1** | (1.06±0.08)E+0 | **(9.97±0.07)E-1** |
| Chaotic | GS | (2.51±0.14)E-1 | **(9.35±0.05)E-2** | **(9.45±0.04)E-2** | (9.55±0.05)E-2 | (9.50±0.05)E-2 | **(9.30±0.04)E-2** |
| | KS | (9.95±0.04)E-1 | **(9.66±0.03)E-1** | (1.00±0.02)E+0 | **(9.98±0.01)E-1** | (1.00±0.02)E+0 | **(9.98±0.01)E-1** |
| High dim | PNd | (5.00±0.40)E-3 | **(1.62±0.21)E-3** | (4.13±0.32)E-3 | **(8.21±0.62)E-4** | (7.83±0.54)E-2 | **(1.61±0.23)E-3** |
| | HNd | (3.24±0.26)E-1 | **(9.21±0.82)E-4** | (5.13±0.34)E-1 | **(3.31±0.22)E-4** | (5.24±0.32)E-1 | **(3.31±0.22)E-4** |
| Inverse | PInv | **(8.12±0.64)E-2** | (8.34±0.71)E-1 | (1.25±0.09)E-1 | **(9.40±0.71)E-2** | (2.52±0.24)E-1 | **(1.31±0.12)E-1** |
| | HInv | (4.91±0.42)E+0 | **(6.80±0.52)E-1** | (5.53±0.43)E-1 | **(5.21±0.42)E-2** | (2.12±0.14)E-1 | **(6.02±0.41)E-2** |

Table 6 demonstrates the dominance of our scheme. The percentage of superiority is 81.8% for *gPINN*, 72.7% for *LAAF* and 90.1% for *GAAF*. Moreover, there is a significant drawdown only for *Burgers 2d-C*.

## B  ANOTHER SPP REFORMULATIONS

In this section, we compare `BGDA` with approaches based on saddle-point reformulation that have been proposed in the literature. Namely, Augmented Lagrangian relaxation method for PINNs (`AL-PINN`) (Son et al., 2023) and `dual-dimer` method (Liu and Wang, 2021). `AL-PINN` reformulates the training of *PINN*s as a constrained optimization problem, where initial and boundary conditions are enforced through constraints rather than just penalty terms, and solves a max-min problem during training. `dual-dimer` introduces weights and and additional maximization similar to our methodology, but in Euclidean geometry.

In Table 7, we provide comparison of the best achieved **L2RE**s for `AL-PINN` and `dual-dimer` with ones provided by our approach. All models are trained to the performance limit. Table 7 demonstrates that our scheme dominates `AL-PINN` and `dual-dimer` in 63.6% and 77.3% of

cases, respectively. The consistent superiority over `dual-dimer` highlights the importance of the non-Euclidean nature of the proposed descent-ascent scheme.

Table 7: Comparison of `AdaptiveBGDA` to the AL-PINN. **L2RE** is used as a quality metric. We highlight the **best** result for each PDE.

| PDE | | Optimizer | | |
|-----|-----|-----|-----|-----|
| | | AL-PINN | dual-dimer | AdaBGDA |
| Burgers | 1d-C | (1.30±0.03)E-2 | **(1.22±0.02)E-2** | (1.31±0.02)E-2 |
| | 2d-C | (4.58±0.08)E-1 | (4.54±0.07)E-1 | **(4.21±0.06)E-1** |
| Poisson | 2d-C | (6.04±0.42)E-1 | (4.11±0.29)E-1 | **(8.19±0.20)E-3** |
| | 2d-CG | (3.98±0.27)E-1 | (7.34±0.51)E-2 | **(1.83±0.11)E-2** |
| | 3d-CG | (2.02±0.13)E-1 | (1.59±0.12)E-1 | **(4.81±0.33)E-2** |
| | 2d-MS | (5.62±0.35)E-1 | (3.74±0.23)E-1 | **(3.47±0.13)E-1** |
| Heat | 2d-VC | **(2.78±0.20)E-1** | (6.02±0.40)E-1 | (2.95±0.23)E-1 |
| | 2d-MS | (9.44±0.70)E-3 | **(8.11±0.63)E-3** | (1.87±0.12)E-2 |
| | 2d-CG | (1.12±0.08)E-2 | (1.13±0.08)E-2 | **(9.87±0.17)E-3** |
| | 2d-LT | (9.97±0.01)E-1 | **(9.95±0.01)E-1** | (9.98±0.01)E-1 |
| NS | 2d-C | **(1.02±0.08)E-2** | (2.33±0.12)E-2 | (2.21±0.13)E-2 |
| | 2d-CG | (1.11±0.08)E-1 | **(6.50±0.46)E-2** | (7.70±0.62)E-2 |
| | 2d-LT | (9.88±0.02)E-1 | (9.85±0.02)E-1 | **(9.74±0.02)E-1** |
| Wave | 1d-C | (2.79±0.22)E-1 | (2.61±0.19)E-1 | **(1.62±0.09)E-2** |
| | 2d-CG | (8.02±0.04)E-1 | (7.99±0.06)E-1 | **(7.81±0.05)E-1** |
| | 2d-MS | (1.00±0.03)E+0 | (1.00±0.03)E+0 | **(8.99±0.05)E-1** |
| Chaotic | GS | **(9.30±0.04)E-2** | (9.32±0.04)E-2 | (9.31±0.04)E-2 |
| | KS | (9.62±0.03)E-1 | (9.72±0.03)E-1 | **(9.52±0.02)E-1** |
| High dim | PNd | **(8.10±0.61)E-5** | (4.09±0.33)E-4 | (1.25±0.09)E-4 |
| | HNd | (3.71±0.23)E-4 | (2.60±0.24)E-4 | **(1.57±0.11)E-4** |
| Inverse | PInv | **(7.31±0.12)E-2** | (7.42±0.11)E-2 | (7.57±0.15)E-2 |
| | HInv | (7.22±0.50)E-1 | (1.05±0.82)E+0 | **(4.08±0.31)E-2** |

## C ROBUSTNESS TO VARIATIONS IN HYPERPARAMETERS

In our work, hyperparameters were selected once by tuning to best convergence on *Poisson 2d-C* from *PINNacle* (Hao et al., 2023). In this section, we study the sensitivity of `AdaptiveBGDA` to the choice of hyperparameters. In this experiment, we use *Burgers 1d-C*. Let us start with varying the descent $\gamma_\theta$ and ascent $\gamma_\pi$ step sizes. Table 8 demonstrates robustness to variations in step sizes. This

Table 8: Robustness of `AdaptiveBGDA` to variations in $\gamma_\theta$, $\gamma_\pi$. **L2RE** is used as a quality metric.

| $\gamma_\theta$ | 0.001 | 0.001 | 0.001 | 0.004 | 0.004 | 0.004 | 0.016 | 0.016 | 0.016 |
|-----|-----|-----|-----|-----|-----|-----|-----|-----|-----|
| $\gamma_\pi$ | 0.01 | 0.1 | 0.5 | 0.01 | 0.1 | 0.5 | 0.01 | 0.1 | 0.5 |
| **L2RE** | **1.26E-2** | 1.30E-2 | 1.28E-2 | 1.30E-2 | 1.31E-2 | 1.31E-2 | 1.31E-2 | 1.30E-2 | 1.35E-2 |

allows to obtain satisfactory results on the benchmark experiments (see Table 3) without additional tuning for each specific PDE. We note that `AdaptiveBGDA` is also robust to poor tuning of $\lambda$.

Table 9: Robustness of `AdaptiveBGDA` to variations in $\lambda$. **L2RE** is used as a quality metric.

| $\lambda$ | 0.001 | 0.005 | 0.01 | 0.05 |
|-----|-----|-----|-----|-----|
| **L2RE** | 1.30E-2 | **1.26E-2** | **1.26E-2** | 1.31E-2 |

## D COMPARISON OF THEORETICAL AND EMPIRICAL RESULTS

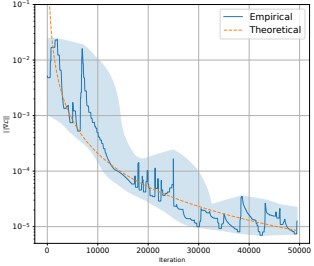

Figure 5: Comparison of theory and practice for `AdaptiveBGDA`

Convergence guarantees of `BGDA` are established under the Lipschitz continuity of $\nabla\mathcal{L}$. Despite the idealized nature of this assumption, it provides a reasonably good approximation of real loss functions (Khromov and Singh). Figure 5 illustrates a comparison between the empirical behavior of our method and the theoretical convergence. Since theory guarantees a decrease in the gradient norm, we measure the convergence of `BGDA` according to this criterion. Since the convergence bound contains constants that cannot be computed in practice ($L$, $\kappa$, $\Phi(\theta^0) - \Phi(\theta^*)$, $D_\psi(\pi^{(\theta^0)}, \pi^0)$), we define the theoretical convergence function as $f(t) = C/\sqrt{t}$ and check whether there exists a constant $C$ such that the plot of $f(t)$ lies within the confidence interval of the curve corresponding to the gradient norm. In Figure 5, $C = 20811$.

On the logarithmic scale, it can be seen that the empirical curve decreases at the same rate as the theoretical reference: the slopes of the lines nearly coincide, and the discrepancy between them remains stable throughout all iterations. This confirms that the actual convergence behavior of `BGDA` aligns with the theoretical predictions, and that the theoretical guarantees adequately reflect its practical dynamics.

We also provide a comparison of the convergence speed of `AdaptiveBGDA` against the competing methods on *Burgers 1d-C*. See Figure 6 for the results.

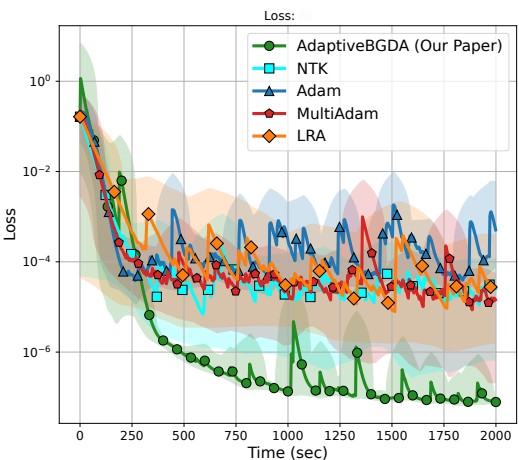

Figure 6: Comparison of `AdaptiveBGDA` to competitors on *Wave 1d-C*. Training MSE loss is used as a metric.

## E  STRONG CONCAVITY OF THE OBJECTIVE

In this section, we prove Lemma 1. It follows obviously from the form of the objective (see 2) and Assumption 2.

**Lemma 3.** (**Lemma 1**). *Consider the problem 2 under Assumption 2. Then, for every $\theta \in \mathbb{R}^d$ the function $\mathcal{L}(\theta, \pi)$ is $\lambda$-**strongly concave**, i.e. for all $\pi_1, \pi_2 \in S$ it satisfies*

$$\mathcal{L}(\theta, \pi_1) \leq \mathcal{L}(\theta, \pi_2) + \langle \nabla_\psi \mathcal{L}(\theta, \pi_2), \pi_1 - \pi_2 \rangle - \frac{\lambda}{2} \left( D_\psi(\pi_1, \pi_2) + D_\psi(\pi_2, \pi_1) \right).$$

*Proof.* Note that $\nabla^2_\pi \mathcal{L}(\theta, \pi) = -\lambda \nabla^2 \psi(\pi)$. The function $\mathcal{L}(\theta, \pi)$ is $\mu$-strongly concave related to $D_\psi$, if $\nabla^2_\pi \mathcal{L}(\theta, \pi) \preceq -\mu \nabla^2 \psi(\pi)$ (Lu et al., 2018). Therefore, the objective is $\lambda$-stongly relatively concave. $\square$

# F    PROOF OF LEMMA 2

We begin the presentation of the analysis with a key result guaranteeing convergence. It demonstrates that the distance between $\pi^t$ and the exact maximum of $\pi^*(\theta^t)$ has a suitable dynamics with increasing $t$.

**Lemma 4.** *(**Lemma 2**). Consider the problem 2 under Assumptions 1, 2. Then, Algorithm 1 with tuning*

$$\gamma_\pi = \frac{\lambda}{4L^2}, \quad \gamma_\theta \leq \frac{1}{184\kappa^4 L}$$

*produces such $\{(\theta^t, \pi^t)\}_{t=1}^T$, that*

$$D_\psi(\pi^*(\theta^{t+1}), \pi^{t+1}) \leq \left(1 - \frac{1}{64\kappa^2}\right) D_\psi(\pi^*(\theta^t), \pi^t) + 264\gamma_\theta^2 \kappa^6 \|\nabla\Phi(\theta^t)\|^2,$$

*where $\kappa = L/\lambda$ is the condition number of $\mathcal{L}(\theta, \pi)$ in $\pi$.*

*Proof.* Before proceeding to the proof, let us recall the three-point identity. It plays a key role in the analysis of Bregman methods.

$$D_\psi(x, y) - D_\psi(x, z) - D_\psi(z, y) = \langle \nabla\psi(z) - \nabla\psi(y), x - z \rangle. \tag{3}$$

To begin, we use equation 3 in the form

$$\begin{aligned}
D_\psi(\pi^*(\theta^{t+1}), \pi^{t+1}) =& D_\psi(\pi^*(\theta^{t+1}), \pi^*(\theta^t)) + D_\psi(\pi^*(\theta^t), \pi^{t+1}) \\
&+ \langle \nabla\psi(\pi^*(\theta^t)) - \nabla\psi(\pi^{t+1}), \pi^*(\theta^{t+1}) - \pi^*(\theta^t) \rangle.
\end{aligned} \tag{4}$$

Further, we write the optimality condition for Line 5:

$$\left\langle -\gamma_\pi \nabla_\pi \mathcal{L}(\theta^t, \pi^t) + [\nabla\psi(\pi^{t+1}) - \nabla\psi(\pi^t)], \pi^*(\theta^t) - \pi^{t+1} \right\rangle \geq 0.$$

Applying equation 3, we obtain

$$-\gamma_\pi \left\langle \nabla_\pi \mathcal{L}(\theta^t, \pi^t), \pi^*(\theta^t) - \pi^{t+1} \right\rangle + D_\psi(\pi^*(\theta^t), \pi^t) - D_\psi(\pi^*(\theta^t), \pi^{t+1}) - D_\psi(\pi^{t+1}, \pi^t) \geq 0.$$

After re-arranging the terms, we get

$$D_\psi(\pi^*(\theta^t), \pi^{t+1}) \leq D_\psi(\pi^*(\theta^t), \pi^t) - D_\psi(\pi^{t+1}, \pi^t) - \gamma_\pi \left\langle \nabla_\pi \mathcal{L}(\theta^t, \pi^t), \pi^*(\theta^t) - \pi^{t+1} \right\rangle. \tag{5}$$

Since $\pi^*(\theta^t)$ is the exact maximum of $\mathcal{L}(\theta^t, \pi)$ in $\pi$, there is another optimility condition

$$\gamma_\pi \left\langle \nabla_\pi \mathcal{L}(\theta^t, \pi^*(\theta^t)), \pi^*(\theta^t) - \pi \right\rangle \geq 0.$$

Substituting $\pi = \pi^{t+1}$ and summing it with equation 5, we derive

$$\begin{aligned}
D_\psi(\pi^*(\theta^t), \pi^{t+1}) \leq& D_\psi(\pi^*(\theta^t), \pi^t) - D_\psi(\pi^{t+1}, \pi^t) \\
&+ \gamma_\pi \left\langle \nabla_\pi \mathcal{L}(\theta^t, \pi^*(\theta^t)) - \nabla_\pi \mathcal{L}(\theta^t, \pi^t), \pi^*(\theta^t) - \pi^{t+1} \right\rangle \\
\leq& D_\psi(\pi^*(\theta^t), \pi^t) - D_\psi(\pi^{t+1}, \pi^t) \\
&+ \gamma_\pi \left\langle \nabla_\pi \mathcal{L}(\theta^t, \pi^*(\theta^t)) - \nabla_\pi \mathcal{L}(\theta^t, \pi^t), \pi^*(\theta^t) - \pi^t \right\rangle \\
&+ \gamma_\pi \left\langle \nabla_\pi \mathcal{L}(\theta^t, \pi^*(\theta^t)) - \nabla_\pi \mathcal{L}(\theta^t, \pi^t), \pi^t - \pi^{t+1} \right\rangle.
\end{aligned}$$

Now, we are going to utilize the strong concavity of $\mathcal{L}(\theta, \pi)$ in $\pi$:

$$\gamma_\pi \left\langle \nabla_\pi \mathcal{L}(\theta^t, \pi^*(\theta^t)) - \nabla_\pi \mathcal{L}(\theta^t, \pi^t), \pi^*(\theta^t) - \pi^t \right\rangle \leq \frac{-\gamma_\pi \lambda}{2} D_\psi(\pi^*(\theta^t), \pi^t).$$

Thus, we have

$$\begin{aligned}
D_\psi(\pi^*(\theta^t), \pi^{t+1}) \leq& \left(1 - \frac{\gamma_\pi \lambda}{2}\right) D_\psi(\pi^*(\theta^t), \pi^t) - D_\psi(\pi^{t+1}, \pi^t) \\
&+ \gamma_\pi \left\langle \nabla_\pi \mathcal{L}(\theta^t, \pi^*(\theta^t)) - \nabla_\pi \mathcal{L}(\theta^t, \pi^t), \pi^t - \pi^{t+1} \right\rangle.
\end{aligned}$$

Next, we apply Cauchy-Schwartz inequality to the scalar product and obtain

$$\begin{aligned}
D_\psi(\pi^*(\theta^t), \pi^{t+1}) \leq& \left(1 - \frac{\gamma_\pi \lambda}{2}\right) D_\psi(\pi^*(\theta^t), \pi^t) - D_\psi(\pi^{t+1}, \pi^t) \\
&+ \frac{\gamma_\pi \alpha}{2} \|\nabla_\pi \mathcal{L}(\theta^t, \pi^*(\theta^t)) - \nabla_\pi \mathcal{L}(\theta^t, \pi^t)\|^2 + \frac{\gamma_\pi}{2\alpha} \|\pi^t - \pi^{t+1}\|^2.
\end{aligned}$$

Using $L$-smoothness of $\mathcal{L}$ (see Assumption 1), we obtain

$$D_\psi(\pi^*(\theta^t), \pi^{t+1}) \leq \left(1 - \frac{\gamma_\pi \lambda}{2}\right) D_\psi(\pi^*(\theta^t), \pi^t) - D_\psi(\pi^{t+1}, \pi^t)$$
$$+ \frac{\gamma_\pi \alpha L^2}{2} \|\pi^*(\theta^t) - \pi^t\|^2 + \frac{\gamma_\pi}{2\alpha} \|\pi^t - \pi^{t+1}\|^2.$$

Since $\psi$ is 1-strongly convex (see Assumption 2), we have

$$\frac{1}{2}\|\pi_1 - \pi_2\|^2 \leq D_\psi(\pi_1, \pi_2).$$

Thus,

$$D_\psi(\pi^*(\theta^t), \pi^{t+1}) \leq \left(1 - \frac{\gamma_\pi \lambda}{2}\right) D_\psi(\pi^*(\theta^t), \pi^t) - D_\psi(\pi^{t+1}, \pi^t)$$
$$+ \gamma_\pi \alpha L^2 D_\psi(\pi^*(\theta^t), \pi^t) + \frac{\gamma_\pi}{\alpha} D_\psi(\pi^t, \pi^{t+1}).$$

Choose $\alpha = \gamma_\pi$. We can derive

$$D_\psi(\pi^*(\theta^t), \pi^{t+1}) \leq \left(1 - \frac{\gamma_\pi \lambda}{2} + \gamma_\pi^2 L^2\right) D_\psi(\pi^*(\theta^t), \pi^t).$$

Since $\gamma_\pi = \lambda/4L^2$, we have

$$D_\psi(\pi^*(\theta^t), \pi^{t+1}) \leq \left(1 - \frac{1}{16\kappa^2}\right) D_\psi(\pi^*(\theta^t), \pi^t). \tag{6}$$

Let us return to equation 4. Note that

$$\nabla\psi(\pi^*(\theta^t)) - \nabla\psi(\pi^{t+1}) = \frac{1}{\lambda}\left(\nabla_\pi \mathcal{L}(\theta^t, \pi^{t+1}) - \nabla_\pi \mathcal{L}(\theta^t, \pi^*(\theta^t))\right).$$

Thus, there is

$$D_\psi(\pi^*(\theta^{t+1}), \pi^{t+1}) = D_\psi(\pi^*(\theta^{t+1}), \pi^*(\theta^t)) + D_\psi(\pi^*(\theta^t), \pi^{t+1})$$
$$+ \frac{1}{\lambda}\langle \nabla_\pi \mathcal{L}(\theta^t, \pi^{t+1}) - \nabla_\pi \mathcal{L}(\theta^t, \pi^*(\theta^t)), \pi^*(\theta^{t+1}) - \pi^*(\theta^t)\rangle$$
$$\leq D_\psi(\pi^*(\theta^{t+1}), \pi^*(\theta^t)) + D_\psi(\pi^*(\theta^t), \pi^{t+1})$$
$$+ \frac{\alpha L^2}{\lambda} D_\psi(\pi^*(\theta^t), \pi^{t+1}) + \frac{1}{\lambda\alpha} D_\psi(\pi^*(\theta^{t+1}), \pi^*(\theta^t)).$$

Let us choose $\alpha = \lambda^3/32L^4$. With such a choice, we have

$$D_\psi(\pi^*(\theta^{t+1}), \pi^{t+1}) \leq 33\kappa^4 D_\psi(\pi^*(\theta^{t+1}), \pi^*(\theta^t)) + \left(1 + \frac{1}{32\kappa^2}\right) D_\psi(\pi^*(\theta^t), \pi^{t+1}).$$

To deal with $D_\psi(\pi^*(\theta^t), \pi^{t+1})$, we utilize equation 6. As a result, we obtain

$$D_\psi(\pi^*(\theta^{t+1}), \pi^{t+1}) \leq 33\kappa^4 D_\psi(\pi^*(\theta^{t+1}), \pi^*(\theta^t)) + \left(1 - \frac{1}{32\kappa^2}\right) D_\psi(\pi^*(\theta^t), \pi^t). \tag{7}$$

The rest thing is to prove that the descent step does not dramatically change the distance between the optimal values of weights. Let us write down two optimality conditions:

$$\langle \nabla_\pi \mathcal{L}(\theta^t, \pi^*(\theta^t)), \pi - \pi^*(\theta^t)\rangle \leq 0,$$
$$\langle \nabla_\pi \mathcal{L}(\theta^{t+1}, \pi^*(\theta^{t+1})), \pi - \pi^*(\theta^{t+1})\rangle \leq 0.$$

Let us substitute $\pi = \pi^*(\theta^{t+1})$ into the first inequality and $\pi = \pi^*(\theta^t)$ into the second one. When summing them up, we have

$$\langle \nabla_\pi \mathcal{L}(\theta^t, \pi^*(\theta^t)) - \nabla_\pi \mathcal{L}(\theta^{t+1}, \pi^*(\theta^{t+1})), \pi^*(\theta^{t+1}) - \pi^*(\theta^t)\rangle \leq 0. \tag{8}$$

On the other hand, we can take advantage of the strong concavity of the objective (see Lemma 1) and write

$$\langle \nabla_\pi \mathcal{L}(\theta^t, \pi^*(\theta^{t+1})) - \nabla_\pi \mathcal{L}(\theta^t, \pi^*(\theta^t)), \pi^*(\theta^{t+1}) - \pi^*(\theta^t)\rangle$$
$$\leq -\frac{\lambda}{2}\left[D_\psi(\pi^*(\theta^t), \pi^*(\theta^{t+1})) + D_\psi(\pi^*(\theta^{t+1}), \pi^*(\theta^t))\right]. \tag{9}$$

Combining equation 8 and equation 9, we obtain

$$\frac{\lambda^2}{4} \left[ D_\psi(\pi^*(\theta^t), \pi^*(\theta^{t+1})) + D_\psi(\pi^*(\theta^{t+1}), \pi^*(\theta^t)) \right]^2 \le L^2 \|\pi^*(\theta^{t+1}) - \pi^*(\theta^t)\|^2 \cdot \|\theta^{t+1} - \theta^t\|^2.$$

Applying the strong convexity of distance generating function (Assumption 2) and re-arranging terms, we obtain

$$D_\psi(\pi^*(\theta^t), \pi^*(\theta^{t+1})) + D_\psi(\pi^*(\theta^{t+1}), \pi^*(\theta^t)) \le 4\kappa^2 \|\theta^{t+1} - \theta^t\|^2 \le 4\gamma_\theta^2 \kappa^2 \|\nabla_\theta \mathcal{L}(\theta^t, \pi^t)\|^2.$$

Next, we ass and subtract $\nabla \Phi(\theta^t)$ and apply Assumption 1. We obtain

$$D_\psi(\pi^*(\theta^t), \pi^*(\theta^{t+1})) + D_\psi(\pi^*(\theta^{t+1}), \pi^*(\theta^t)) \le 16\gamma_\theta^2 \kappa^2 L^2 D_\psi(\pi^*(\theta^t), \pi^t) + 8\gamma_\theta^2 \kappa^2 \|\nabla \Phi(\theta^t)\|^2.$$

Thus, equation 7 transforms into

$$D_\psi(\pi^*(\theta^{t+1}), \pi^{t+1}) \le \left( 1 - \frac{1}{32\kappa^2} + 528\gamma_\theta^2 \kappa^6 L^2 \right) D_\psi(\pi^*(\theta^t), \pi^t) + 264\gamma_\theta^2 \kappa^6 \|\nabla \Phi(\theta^t)\|^2.$$

With $\gamma_\theta \le {}^1/_{184\kappa^4 L}$, we obtain

$$D_\psi(\pi^*(\theta^{t+1}), \pi^{t+1}) \le \left( 1 - \frac{1}{64\kappa^2} \right) D_\psi(\pi^*(\theta^t), \pi^t) + 264\gamma_\theta^2 \kappa^6 \|\nabla \Phi(\theta^t)\|^2.$$

This completes the proof. $\qquad\square$

## G   PROOF OF THEOREM 1

**Theorem 2.** *(Theorem 1) Consider the problem 2 under Assumptions 1, 2. Then, Algorithm 1 with tuning*

$$\gamma_\pi = \frac{\lambda}{4L^2}, \quad \gamma_\theta \le \sqrt{\frac{43}{92 * 33792}} \frac{1}{\kappa^4 L}$$

*requires*

$$\mathcal{O}\left( \frac{\kappa^4 L \Delta + \kappa^2 L^2 D_\psi(\pi^*(\theta^0), \pi^0)}{\varepsilon^2} \right) \text{ iterations}$$

*to achieve an arbitrary $\varepsilon$-solution, where $\varepsilon^2 = \frac{1}{T} \sum_{t=1}^{T-1} \|\nabla \Phi(\theta^t)\|^2$, $\Delta = \Phi(\theta^0) - \Phi(\theta^*)$. $\kappa = {}^L/_\lambda$.*

*Proof.* One can note that $\Phi$ is $3\kappa L$-smooth. Indeed,

$$\begin{aligned}
\|\nabla \Phi(\theta_1) - \nabla \Phi(\theta_2)\|^2 &= \|\nabla_\theta \mathcal{L}(\theta_1, \pi^*(\theta_1)) - \nabla_\theta \mathcal{L}(\theta_2, \pi^*(\theta_2))\|^2 \\
&\le L^2 \left[ \|\theta_1 - \theta_2\|^2 + 2D_\psi(\pi^*(\theta_1), \pi^*(\theta_2)) \right] \le L^2 \left( 1 + 4\kappa^2 \right) \|\theta_1 - \theta_2\|^2 \\
&\le 9\kappa^2 L^2 \|\theta_1 - \theta_2\|^2.
\end{aligned}$$

Thus, we can write

$$\begin{aligned}
\Phi(\theta^{t+1}) &\le \Phi(\theta^t) + \langle \nabla \Phi(\theta^t), \theta^{t+1} - \theta^t \rangle + 3\kappa L \|\theta^{t+1} - \theta^t\|^2 \\
&\le \Phi(\theta^t) - \gamma_\theta \|\nabla \Phi(\theta^t)\|^2 + 3\gamma_\theta^2 \kappa L \|\nabla_\theta \mathcal{L}(\theta^t, \pi^t)\|^2 \\
&\quad + \gamma_\theta \langle \nabla \Phi(\theta^t) - \nabla_\theta \mathcal{L}(\theta^t, \pi^t), \nabla \Phi(\theta^t) \rangle \\
&\le \Phi(\theta^t) - \frac{\gamma_\theta}{2} \|\nabla \Phi(\theta^t)\|^2 + 3\gamma_\theta^2 \kappa L \|\nabla_\theta \mathcal{L}(\theta^t, \pi^t)\|^2 + \frac{\gamma_\theta}{2} \|\nabla \Phi(\theta^t) - \nabla_\theta \mathcal{L}(\theta^t, \pi^t)\|^2 \\
&\le \Phi(\theta^t) - \left( \frac{\gamma_\theta}{2} - 6\gamma_\theta^2 \kappa L \right) \|\nabla \Phi(\theta^t)\|^2 + \left( \frac{\gamma_\theta}{2} + 6\gamma_\theta^2 \kappa L \right) \|\nabla \Phi(\theta^t) - \nabla_\theta \mathcal{L}(\theta^t, \pi^t)\|^2.
\end{aligned}$$

Note that

$$- \left( \frac{\gamma_\theta}{2} - 6\gamma_\theta^2 \kappa L \right) \le - \frac{43\gamma_\theta}{92}.$$

On the other hand,

$$\left( \frac{\gamma_\theta}{2} + 6\gamma_\theta^2 \kappa L \right) \le \gamma_\theta.$$

Thus, we have

$$\begin{aligned}
\Phi(\theta^{t+1}) &\le \Phi(\theta^t) - \frac{43\gamma_\theta}{92} \|\nabla \Phi(\theta^t)\|^2 + \gamma_\theta \|\nabla \Phi(\theta^t) - \nabla_\theta \mathcal{L}(\theta^t, \pi^t)\|^2 \\
&\le \Phi(\theta^t) - \frac{43\gamma_\theta}{92} \|\nabla \Phi(\theta^t)\|^2 + 2\gamma_\theta L^2 D_\psi(\pi^*(\theta^t), \pi^t).
\end{aligned}$$

Let us denote $\delta = 1 - 1/64\kappa^2$. Lemma 2 transforms into

$$D_\psi(\pi^*(\theta^t), \pi^t) \le \delta^t D_\psi(\pi^*(\theta^0), \pi^0) + 264\gamma_\theta^2 \kappa^6 \sum_{j=0}^{t-1} \delta^{t-1-j} \|\nabla\Phi(\theta^j)\|^2.$$

Hence,

$$\Phi(\theta^{t+1}) \le \Phi(\theta^t) - \frac{43\gamma_\theta}{92} \|\nabla\Phi(\theta^t)\|^2 + 2\gamma_\theta L^2 \delta^t D_\psi(\pi^*(\theta^0), \pi^0)$$
$$+ 528\gamma_\theta^3 \kappa^6 L^2 \sum_{j=0}^{t-1} \delta^{t-1-j} \|\nabla\Phi(\theta^j)\|^2.$$

Let us sum up over the iterates $t$ and obtain

$$\Phi(\theta^T) \le \Phi(\theta^0) - \frac{43\gamma_\theta}{92} \sum_{t=1}^{T-1} \|\nabla\Phi(\theta^t)\|^2 + 2\gamma_\theta L^2 \sum_{t=1}^{T-1} \delta^t D_\psi(\pi^*(\theta^0), \pi^0)$$
$$+ 528\gamma_\theta^3 \kappa^6 L^2 \sum_{t=1}^{T-1} \sum_{j=0}^{t-1} \delta^{t-1-j} \|\nabla\Phi(\theta^j)\|^2.$$

Next, we use the property of geometric progression and write

$$\Phi(\theta^T) \le \Phi(\theta^0) - \frac{43\gamma_\theta}{92} \sum_{t=1}^{T-1} \|\nabla\Phi(\theta^t)\|^2 + 128\gamma_\theta \kappa^2 L^2 D_\psi(\pi^*(\theta^0), \pi^0)$$
$$+ 33792\gamma_\theta^3 \kappa^8 L^2 \sum_{t=1}^{T-1} \|\nabla\Phi(\theta^t)\|^2.$$

Choosing $\gamma_\theta \le \sqrt{\frac{43}{92*33792}} \frac{1}{\kappa^4 L}$. Thus, we derive

$$\frac{1}{T} \sum_{t=1}^{T-1} \|\nabla\Phi(\theta^t)\|^2 \le \mathcal{O}\left( \frac{\kappa^4 L \Delta_\Phi}{T} + \frac{\kappa^2 L^2 D_\psi(\pi^*(\theta^0), \pi^0)}{T} \right).$$

$\square$

## H   ENHANCED RATES ON REGULARIZED SIMPLEX

The theory presented in Appendices F, G is constructed for and arbitrary Bregman divergence. This is the main reason for the deterioration of the theoretical guarantees compared to the Euclidean setting. In this section, we look towards the selection of the efficient approach for determining the set of weights $S$. We consider a classic approach of using a unit simplex $\triangle_1^{M-1}$:

$$\triangle_1^{M-1} = \left\{ (\pi_1, \ldots, \pi_M) : \pi_m \ge 0, \sum_{m=1}^{M} \pi_m = 1, \right\}.$$

Note that $\psi(\pi) = -\sum_{m=1}^{M} \pi_m \log \pi_m$ goes to infinity at vertices of $\triangle_1^{M-1}$. Thus, one cannot guarantee smoothness of $\mathcal{L}(\theta, \pi)$ in $\pi$ for every fixed $\theta$. To avoid this, we propose to intersect the simplex by a euclidean ball. This approach is common in the literature (Mehta et al., 2024). Thus, we deal with

$$S = \triangle_1^{M-1} \cap B_{\|\cdot\|}(\mathcal{U}, R),$$

where $\mathcal{U} = (1/M, \ldots, 1/M)^\top$.

**Lemma 5.** *The function $\mathcal{L}(\theta, \pi)$ is $L_\pi$-smooth in $\pi$, i.e. for all $\pi_1, \pi_2 \in S$ it satisfies*
$$\|\nabla\mathcal{L}(\theta, \pi_1) - \nabla\mathcal{L}(\theta, \pi_2)\| \le L_\pi \|\pi_1 - \pi_2\|^2.$$
*Moreover, under strong regularization ($R \ll 1$), it is*
$$L_\pi = \Theta(\lambda M^2 R).$$

*Proof.* Without loss of generality, consider $\pi = (a, b, \dots, b)$, where $a = \min_m \pi_m$. Note that

$$\|\nabla_\pi^2 \mathcal{L}(\theta, \pi)\| = \lambda \left\| \text{diag}\left(\frac{1}{\pi_1}, \dots, \frac{1}{\pi_M}\right) \right\|.$$

Thus, we need to find $\max_{a \in \triangle_1^{M-1}} \frac{1}{a}$ with $\|\pi - \mathcal{U}\|^2 \le R^2$. Let us write

$$\|\pi - \mathcal{U}\|^2 = \left(a - \frac{1}{M}\right)^2 + (M - 1)\left(b - \frac{1}{M}\right)^2 \le R^2. \tag{10}$$

Consider $b = \frac{1-a}{M-1}$. Then equation 10 transforms into

$$\left(a - \frac{1}{M}\right)^2 + \frac{(1 - aM)^2}{M^2(M-1)} \le R^2.$$

Solving the one-dimensional optimization problem, we find the Lipschitz constant of $\nabla_\pi \mathcal{L}(\theta, \pi)$. If $R \ll 1$, then

$$L_\pi = \frac{\lambda}{1/M - \Theta(R)} = \frac{\lambda M}{1 - M\Theta(R)} \approx \Theta(\lambda M^2 R).$$

$\square$

Note that this value is negligible. Indeed, $R \in (0, 1)$, and $M$ in problems of mathematical physics (see equation 1) is usually equal to 3–4. Thus, if $\kappa_\pi = {L_\pi}/{\lambda}$ appears in the estimate, it is comparable in magnitude to other constants hidden in the big-O.

Now let us move to an analysis with enhanced rate.

**Lemma 6.** *Consider the problem 2 under Assumptions 1, 2. Let $S = \triangle_1^{M-1} \cap B_{\|\cdot\|}(\mathcal{U}, R)$. Then, Algorithm 1 with tuning*

$$\gamma_\pi = \frac{\lambda}{4L_\pi^2}, \quad \gamma_\theta \le \frac{1}{184\kappa_\pi^3 \kappa L}$$

*produces such $\{(\theta^t, \pi^t)\}_{t=1}^T$, that*

$$D_\psi(\pi^*(\theta^{t+1}), \pi^{t+1}) \le \left(1 - \frac{1}{64\kappa_\pi^2}\right) D_\psi(\pi^*(\theta^t), \pi^t) + 264\gamma_\theta^2 \kappa_\pi^4 \kappa^2 \|\nabla\Phi(\theta^t)\|^2,$$

*where $\kappa = {L}/{\lambda}$, $\kappa_\pi = {L_\pi}/{\lambda}$.*

*Proof.* To begin, we use equation 3 in the form

$$D_\psi(\pi^*(\theta^{t+1}), \pi^{t+1}) = D_\psi(\pi^*(\theta^{t+1}), \pi^*(\theta^t)) + D_\psi(\pi^*(\theta^t), \pi^{t+1}) \\ + \langle \nabla\psi(\pi^*(\theta^t)) - \nabla\psi(\pi^{t+1}), \pi^*(\theta^{t+1}) - \pi^*(\theta^t) \rangle. \tag{11}$$

Further, we write the optimality condition for Line 5:

$$\left\langle -\gamma_\pi \nabla_\pi \mathcal{L}(\theta^t, \pi^t) + [\nabla\psi(\pi^{t+1}) - \nabla\psi(\pi^t)], \pi^*(\theta^t) - \pi^{t+1} \right\rangle \ge 0.$$

Applying equation 3, we obtain

$$-\gamma_\pi \left\langle \nabla_\pi \mathcal{L}(\theta^t, \pi^t), \pi^*(\theta^t) - \pi^{t+1} \right\rangle + D_\psi(\pi^*(\theta^t), \pi^t) - D_\psi(\pi^*(\theta^t), \pi^{t+1}) - D_\psi(\pi^{t+1}, \pi^t) \ge 0.$$

After re-arranging the terms, we get

$$D_\psi(\pi^*(\theta^t), \pi^{t+1}) \le D_\psi(\pi^*(\theta^t), \pi^t) - D_\psi(\pi^{t+1}, \pi^t) - \gamma_\pi \left\langle \nabla_\pi \mathcal{L}(\theta^t, \pi^t), \pi^*(\theta^t) - \pi^{t+1} \right\rangle. \tag{12}$$

Since $\pi^*(\theta^t)$ is the exact maximum of $\mathcal{L}(\theta^t, \pi)$ in $\pi$, there is another optimility condition

$$\gamma_\pi \left\langle \nabla_\pi \mathcal{L}(\theta^t, \pi^*(\theta^t)), \pi^*(\theta^t) - \pi \right\rangle \ge 0.$$

Substituting $\pi = \pi^{t+1}$ and summing it with equation 12, we derive

$$D_\psi(\pi^*(\theta^t), \pi^{t+1}) \le D_\psi(\pi^*(\theta^t), \pi^t) - D_\psi(\pi^{t+1}, \pi^t) \\ + \gamma_\pi \left\langle \nabla_\pi \mathcal{L}(\theta^t, \pi^*(\theta^t)) - \nabla_\pi \mathcal{L}(\theta^t, \pi^t), \pi^*(\theta^t) - \pi^{t+1} \right\rangle \\ \le D_\psi(\pi^*(\theta^t), \pi^t) - D_\psi(\pi^{t+1}, \pi^t) \\ + \gamma_\pi \left\langle \nabla_\pi \mathcal{L}(\theta^t, \pi^*(\theta^t)) - \nabla_\pi \mathcal{L}(\theta^t, \pi^t), \pi^*(\theta^t) - \pi^t \right\rangle \\ + \gamma_\pi \left\langle \nabla_\pi \mathcal{L}(\theta^t, \pi^*(\theta^t)) - \nabla_\pi \mathcal{L}(\theta^t, \pi^t), \pi^t - \pi^{t+1} \right\rangle.$$

Now, we are going to utilize the strong concavity of $\mathcal{L}(\theta, \pi)$ in $\pi$:

$$\gamma_\pi \left\langle \nabla_\pi \mathcal{L}(\theta^t, \pi^*(\theta^t)) - \nabla_\pi \mathcal{L}(\theta^t, \pi^t), \pi^*(\theta^t) - \pi^t \right\rangle \leq \frac{-\gamma_\pi \lambda}{2} D_\psi(\pi^*(\theta^t), \pi^t).$$

Thus, we have

$$D_\psi(\pi^*(\theta^t), \pi^{t+1}) \leq \left(1 - \frac{\gamma_\pi \lambda}{2}\right) D_\psi(\pi^*(\theta^t), \pi^t) - D_\psi(\pi^{t+1}, \pi^t)$$
$$+ \gamma_\pi \left\langle \nabla_\pi \mathcal{L}(\theta^t, \pi^*(\theta^t)) - \nabla_\pi \mathcal{L}(\theta^t, \pi^t), \pi^t - \pi^{t+1} \right\rangle.$$

Next, we apply Cauchy-Schwartz inequality to the scalar product and obtain

$$D_\psi(\pi^*(\theta^t), \pi^{t+1}) \leq \left(1 - \frac{\gamma_\pi \lambda}{2}\right) D_\psi(\pi^*(\theta^t), \pi^t) - D_\psi(\pi^{t+1}, \pi^t)$$
$$+ \frac{\gamma_\pi \alpha}{2} \|\nabla_\pi \mathcal{L}(\theta^t, \pi^*(\theta^t)) - \nabla_\pi \mathcal{L}(\theta^t, \pi^t)\|^2 + \frac{\gamma_\pi}{2\alpha} \|\pi^t - \pi^{t+1}\|^2.$$

Using $L_\pi$-smoothness of $\mathcal{L}(\theta, \pi)$ in $\pi$ (see Lemma 5), we obtain

$$D_\psi(\pi^*(\theta^t), \pi^{t+1}) \leq \left(1 - \frac{\gamma_\pi \lambda}{2}\right) D_\psi(\pi^*(\theta^t), \pi^t) - D_\psi(\pi^{t+1}, \pi^t)$$
$$+ \frac{\gamma_\pi \alpha L_\pi^2}{2} \|\pi^*(\theta^t) - \pi^t\|^2 + \frac{\gamma_\pi}{2\alpha} \|\pi^t - \pi^{t+1}\|^2.$$

Since $\psi$ is 1-strongly convex (see Assumption 2), we have

$$\frac{1}{2} \|\pi_1 - \pi_2\|^2 \leq D_\psi(\pi_1, \pi_2).$$

Thus,

$$D_\psi(\pi^*(\theta^t), \pi^{t+1}) \leq \left(1 - \frac{\gamma_\pi \lambda}{2}\right) D_\psi(\pi^*(\theta^t), \pi^t) - D_\psi(\pi^{t+1}, \pi^t)$$
$$+ \gamma_\pi \alpha L_\pi^2 D_\psi(\pi^*(\theta^t), \pi^t) + \frac{\gamma_\pi}{\alpha} D_\psi(\pi^t, \pi^{t+1}).$$

Choose $\alpha = \gamma_\pi$. We can derive

$$D_\psi(\pi^*(\theta^t), \pi^{t+1}) \leq \left(1 - \frac{\gamma_\pi \lambda}{2} + \gamma_\pi^2 L_\pi^2\right) D_\psi(\pi^*(\theta^t), \pi^t).$$

Since $\gamma_\pi = \lambda/4L_\pi^2$, we have

$$D_\psi(\pi^*(\theta^t), \pi^{t+1}) \leq \left(1 - \frac{1}{16\kappa_\pi^2}\right) D_\psi(\pi^*(\theta^t), \pi^t). \tag{13}$$

Let us return to equation 11. Note that

$$\nabla \psi(\pi^*(\theta^t)) - \nabla \psi(\pi^{t+1}) = \frac{1}{\lambda} \left(\nabla_\pi \mathcal{L}(\theta^t, \pi^{t+1}) - \nabla_\pi \mathcal{L}(\theta^t, \pi^*(\theta^t))\right).$$

Thus, there is

$$D_\psi(\pi^*(\theta^{t+1}), \pi^{t+1}) = D_\psi(\pi^*(\theta^{t+1}), \pi^*(\theta^t)) + D_\psi(\pi^*(\theta^t), \pi^{t+1})$$
$$+ \frac{1}{\lambda} \langle \nabla_\pi \mathcal{L}(\theta^t, \pi^{t+1}) - \nabla_\pi \mathcal{L}(\theta^t, \pi^*(\theta^t)), \pi^*(\theta^{t+1}) - \pi^*(\theta^t) \rangle$$
$$\leq D_\psi(\pi^*(\theta^{t+1}), \pi^*(\theta^t)) + D_\psi(\pi^*(\theta^t), \pi^{t+1})$$
$$+ \frac{\alpha L_\pi^2}{\lambda} D_\psi(\pi^*(\theta^t), \pi^{t+1}) + \frac{1}{\lambda \alpha} D_\psi(\pi^*(\theta^{t+1}), \pi^*(\theta^t)).$$

Let us choose $\alpha = \lambda^3/32L_\pi^4$. With such a choice, we have

$$D_\psi(\pi^*(\theta^{t+1}), \pi^{t+1}) \leq 33\kappa_\pi^4 D_\psi(\pi^*(\theta^{t+1}), \pi^*(\theta^t)) + \left(1 + \frac{1}{32\kappa_\pi^2}\right) D_\psi(\pi^*(\theta^t), \pi^{t+1}).$$

To deal with $D_\psi(\pi^*(\theta^t), \pi^{t+1})$, we utilize equation 13. As a result, we obtain

$$D_\psi(\pi^*(\theta^{t+1}), \pi^{t+1}) \leq 33\kappa_\pi^4 D_\psi(\pi^*(\theta^{t+1}), \pi^*(\theta^t)) + \left(1 - \frac{1}{32\kappa_\pi^2}\right) D_\psi(\pi^*(\theta^t), \pi^t). \tag{14}$$

The rest thing is to prove that the descent step does not dramatically change the distance between the optimal values of weights. Let us write down two optimality conditions:

$$\langle \nabla_\pi \mathcal{L}(\theta^t, \pi^*(\theta^t)), \pi - \pi^*(\theta^t) \rangle \leq 0,$$
$$\langle \nabla_\pi \mathcal{L}(\theta^{t+1}, \pi^*(\theta^{t+1})), \pi - \pi^*(\theta^{t+1}) \rangle \leq 0.$$

Let us substitute $\pi = \pi^*(\theta^{t+1})$ into the first inequality and $\pi = \pi^*(\theta^t)$ into the second one. When summing them up, we have

$$\langle \nabla_\pi \mathcal{L}(\theta^t, \pi^*(\theta^t)) - \nabla_\pi \mathcal{L}(\theta^{t+1}, \pi^*(\theta^{t+1})), \pi^*(\theta^{t+1}) - \pi^*(\theta^t) \rangle \leq 0. \tag{15}$$

On the other hand, we can take advantage of the strong concavity of the objective (see Lemma 1) and write

$$\langle \nabla_\pi \mathcal{L}(\theta^t, \pi^*(\theta^{t+1})) - \nabla_\pi \mathcal{L}(\theta^t, \pi^*(\theta^t)), \pi^*(\theta^{t+1}) - \pi^*(\theta^t) \rangle$$
$$\leq -\frac{\lambda}{2} \left[ D_\psi(\pi^*(\theta^t), \pi^*(\theta^{t+1})) + D_\psi(\pi^*(\theta^{t+1}), \pi^*(\theta^t)) \right]. \tag{16}$$

Combining equation 15 and equation 16, we obtain

$$\frac{\lambda^2}{4} \left[ D_\psi(\pi^*(\theta^t), \pi^*(\theta^{t+1})) + D_\psi(\pi^*(\theta^{t+1}), \pi^*(\theta^t)) \right]^2 \leq L^2 \|\pi^*(\theta^{t+1}) - \pi^*(\theta^t)\|^2 \cdot \|\theta^{t+1} - \theta^t\|^2.$$

Here we can not apply the smoothness in $\pi$. Instead, we have to use the smoothness in $(\theta, \pi)$. Next, applying the strong convexity of distance generating function (Assumption 2) and re-arranging terms, we obtain

$$D_\psi(\pi^*(\theta^t), \pi^*(\theta^{t+1})) + D_\psi(\pi^*(\theta^{t+1}), \pi^*(\theta^t)) \leq 4\kappa^2 \|\theta^{t+1} - \theta^t\|^2 \leq 4\gamma_\theta^2 \kappa^2 \|\nabla_\theta \mathcal{L}(\theta^t, \pi^t)\|^2.$$

Next, we ass and subtract $\nabla \Phi(\theta^t)$ and apply Assumption 1. We obtain

$$D_\psi(\pi^*(\theta^t), \pi^*(\theta^{t+1})) + D_\psi(\pi^*(\theta^{t+1}), \pi^*(\theta^t)) \leq 16\gamma_\theta^2 \kappa^2 L^2 D_\psi(\pi^*(\theta^t), \pi^t) + 8\gamma_\theta^2 \kappa^2 \|\nabla \Phi(\theta^t)\|^2.$$

Thus, equation 14 transforms into

$$D_\psi(\pi^*(\theta^{t+1}), \pi^{t+1}) \leq \left( 1 - \frac{1}{32\kappa_\pi^2} + 528\gamma_\theta^2 \kappa_\pi^4 \kappa^2 L^2 \right) D_\psi(\pi^*(\theta^t), \pi^t) + 264\gamma_\theta^2 \kappa_\pi^4 \kappa^2 \|\nabla \Phi(\theta^t)\|^2.$$

With $\gamma_\theta \leq 1/184\kappa^3\kappa L$, we obtain

$$D_\psi(\pi^*(\theta^{t+1}), \pi^{t+1}) \leq \left( 1 - \frac{1}{64\kappa_\pi^2} \right) D_\psi(\pi^*(\theta^t), \pi^t) + 264\gamma_\theta^2 \kappa_\pi^4 \kappa^2 \|\nabla \Phi(\theta^t)\|^2.$$

This completes the proof. $\qquad\square$

Next, we modify the main proof to obtain enhanced convergence.

**Theorem 3.** . *Consider the problem 2 under Assumptions 1, 2. Let $S = S = \triangle_1^{M-1} \cap B_{\|\cdot\|}(\mathcal{U}, R)$. Then, Algorithm 1 with tuning*

$$\gamma_\pi = \frac{\lambda}{4L_\pi^2}, \quad \gamma_\theta \leq \sqrt{\frac{43}{92 * 33792}} \frac{1}{\kappa_\pi^3 \kappa L}$$

*requires*

$$\mathcal{O}\left( \frac{\kappa L \Delta + L^2 D_\psi(\pi^*(\theta^0), \pi^0)}{\varepsilon^2} \right) \text{ iterations}$$

*to achieve an arbitrary $\varepsilon$-solution, where $\varepsilon^2 = \frac{1}{T} \sum_{t=1}^{T-1} \|\nabla \Phi(\theta^t)\|^2$, $\Delta = \Phi(\theta^0) - \Phi(\theta^*)$. $\kappa = L/\lambda$, $\kappa_\pi = L_\pi/\lambda$.*

*Proof.* One can note that $\Phi$ is $3\kappa L$-smooth. Indeed,

$$\|\nabla \Phi(\theta_1) - \nabla \Phi(\theta_2)\|^2 = \|\nabla_\theta \mathcal{L}(\theta_1, \pi^*(\theta_1)) - \nabla_\theta \mathcal{L}(\theta_2, \pi^*(\theta_2))\|^2$$
$$\leq L^2 \left[ \|\theta_1 - \theta_2\|^2 + 2D_\psi(\pi^*(\theta_1), \pi^*(\theta_2)) \right] \leq L^2 \left( 1 + 4\kappa^2 \right) \|\theta_1 - \theta_2\|^2$$
$$\leq 9\kappa^2 L^2 \|\theta_1 - \theta_2\|^2.$$

Thus, we can write

$$
\begin{aligned}
\Phi(\theta^{t+1}) \leq & \Phi(\theta^t) + \langle \nabla\Phi(\theta^t), \theta^{t+1} - \theta^t \rangle + 3\kappa L \|\theta^{t+1} - \theta^t\|^2 \\
\leq & \Phi(\theta^t) - \gamma_\theta \|\nabla\Phi(\theta^t)\|^2 + 3\gamma_\theta^2 \kappa L \|\nabla_\theta\mathcal{L}(\theta^t, \pi^t)\|^2 \\
& + \gamma_\theta \langle \nabla\Phi(\theta^t) - \nabla_\theta\mathcal{L}(\theta^t, \pi^t), \nabla\Phi(\theta^t) \rangle \\
\leq & \Phi(\theta^t) - \frac{\gamma_\theta}{2}\|\nabla\Phi(\theta^t)\|^2 + 3\gamma_\theta^2\kappa L\|\nabla_\theta\mathcal{L}(\theta^t, \pi^t)\|^2 + \frac{\gamma_\theta}{2}\|\nabla\Phi(\theta^t) - \nabla_\theta\mathcal{L}(\theta^t, \pi^t)\|^2 \\
\leq & \Phi(\theta^t) - \left(\frac{\gamma_\theta}{2} - 6\gamma_\theta^2\kappa L\right)\|\nabla\Phi(\theta^t)\|^2 + \left(\frac{\gamma_\theta}{2} + 6\gamma_\theta^2\kappa L\right)\|\nabla\Phi(\theta^t) - \nabla_\theta\mathcal{L}(\theta^t, \pi^t)\|^2.
\end{aligned}
$$

Note that

$$
-\left(\frac{\gamma_\theta}{2} - 6\gamma_\theta^2\kappa L\right) \leq -\frac{43\gamma_\theta}{92}.
$$

On the other hand,

$$
\left(\frac{\gamma_\theta}{2} + 6\gamma_\theta^2\kappa L\right) \leq \gamma_\theta.
$$

Thus, we have

$$
\begin{aligned}
\Phi(\theta^{t+1}) \leq & \Phi(\theta^t) - \frac{43\gamma_\theta}{92}\|\nabla\Phi(\theta^t)\|^2 + \gamma_\theta\|\nabla\Phi(\theta^t) - \nabla_\theta\mathcal{L}(\theta^t, \pi^t)\|^2 \\
\leq & \Phi(\theta^t) - \frac{43\gamma_\theta}{92}\|\nabla\Phi(\theta^t)\|^2 + 2\gamma_\theta L^2 D_\psi(\pi^*(\theta^t), \pi^t).
\end{aligned}
$$

Let us denote $\delta = 1 - 1/64\kappa_\pi^2$. Lemma 6 transforms into

$$
D_\psi(\pi^*(\theta^t), \pi^t) \leq \delta^t D_\psi(\pi^*(\theta^0), \pi^0) + 264\gamma_\theta^2\kappa_\pi^4\kappa^2\sum_{j=0}^{t-1}\delta^{t-1-j}\|\nabla\Phi(\theta^j)\|^2.
$$

Hence,

$$
\begin{aligned}
\Phi(\theta^{t+1}) \leq & \Phi(\theta^t) - \frac{43\gamma_\theta}{92}\|\nabla\Phi(\theta^t)\|^2 + 2\gamma_\theta L^2\delta^t D_\psi(\pi^*(\theta^0), \pi^0) \\
& + 528\gamma_\theta^3\kappa_\pi^4\kappa^2 L^2\sum_{j=0}^{t-1}\delta^{t-1-j}\|\nabla\Phi(\theta^j)\|^2.
\end{aligned}
$$

Let us sum up over the iterates $t$ and obtain

$$
\begin{aligned}
\Phi(\theta^T) \leq & \Phi(\theta^0) - \frac{43\gamma_\theta}{92}\sum_{t=1}^{T-1}\|\nabla\Phi(\theta^t)\|^2 + 2\gamma_\theta L^2\sum_{t=1}^{T-1}\delta^t D_\psi(\pi^*(\theta^0), \pi^0) \\
& + 528\gamma_\theta^3\kappa_\pi^4\kappa^2 L^2\sum_{t=1}^{T-1}\sum_{j=0}^{t-1}\delta^{t-1-j}\|\nabla\Phi(\theta^j)\|^2.
\end{aligned}
$$

Next, we use the property of geometric progression and write

$$
\begin{aligned}
\Phi(\theta^T) \leq & \Phi(\theta^0) - \frac{43\gamma_\theta}{92}\sum_{t=1}^{T-1}\|\nabla\Phi(\theta^t)\|^2 + 128\gamma_\theta\kappa_\pi^2 L^2 D_\psi(\pi^*(\theta^0), \pi^0) \\
& + 33792\gamma_\theta^3\kappa_\pi^6\kappa^2 L^2\sum_{t=1}^{T-1}\|\nabla\Phi(\theta^t)\|^2.
\end{aligned}
$$

Choosing $\gamma_\theta \leq \sqrt{\frac{43}{92*33792}}\frac{1}{\kappa_\pi^3\kappa=L}$. Thus, we derive

$$
\frac{1}{T}\sum_{t=1}^{T-1}\|\nabla\Phi(\theta^t)\|^2 \leq \mathcal{O}\left(\frac{\kappa_\pi^3\kappa L\Delta_\Phi}{T} + \frac{\kappa_\pi^2 L^2 D_\psi(\pi^*(\theta^0), \pi^0)}{T}\right).
$$

Above we discussed that $\kappa_\pi$ is small, since not many equations appear in the PDEs systems. Thus, we can focus on $\kappa$ only and proceed to

$$
\frac{1}{T}\sum_{t=1}^{T-1}\|\nabla\Phi(\theta^t)\|^2 \leq \mathcal{O}\left(\frac{\kappa L\Delta_\Phi}{T} + \frac{L^2 D_\psi(\pi^*(\theta^0), \pi^0)}{T}\right).
$$

This finishes the proof. $\qquad\square$

## I  STOCHASTIC SETTING

In the current realities of machine learning, it is almost never possible to use all the data to compute a gradient. Motivated by this fact, we develop a stochastic theory for our scheme. Note that the computation $\nabla_\pi \mathcal{L}(\theta, \pi)$ does not need to perform backward. Therefore, we analyze the stochasticity in $\theta$ only. Consider a stochastic gradient $G_\theta(\theta^t, \pi^t, \xi)$ calculated from one randomly selected sample $\xi$.

**Assumption 3.** *Stochastic oracle $G_\theta$ is unbiased and light-tailed, i.e.*

$$\mathbb{E}_\xi \left[ G_\theta(\theta, \pi, \xi) \right] = \nabla_\theta \mathcal{L}(\theta, \pi), \ \mathbb{E} \left[ \| G_\theta(\theta, \pi, \xi) - \nabla_\theta \mathcal{L}(\theta, \pi) \|^2 \right] \leq \sigma^2, \ \forall (\theta, \pi) \in \mathbb{R}^d \times S.$$

In our analysis, we rely on batching. Namely, we sample a subset of data points and use it to approximate the gradient. The main difference between Algorithm 3 and deterministic BGDA is the

---

**Algorithm 3** S-BGDA

1: **Input:** Starting point $(\theta^0, \pi^0) \in \mathbb{R}^d \times S$, number of iterations $T$
2: **Parameters:** Stepsizes $\gamma_\theta, \gamma_\pi > 0$
3: **for** $t = 0, \dots, T-1$ **do**
4:     Draw a collection of i.i.d. data points $\{\xi_i^t\}_{i=1}^B$
5:     $\theta^{t+1} = \theta^t - \gamma_\theta \frac{1}{B} \sum_{i=1}^B G_\theta(\theta^t, \pi^t, \xi_i^t)$ *// Optimizer updates parameters*
6:     $\pi^{t+1} = \arg\min_{\pi \in S} \left\{ -\gamma_\pi \langle \nabla_\pi \mathcal{L}(\theta^t, \pi^t), \pi \rangle + D_\psi(\pi, \pi^t) \right\}$ *// Optimizer updates weights*
7: **end for**
8: **Output:** $(\theta^T, \pi^T)$

---

use of stochastic oracle call in Line 5.

**Lemma 7.** *Consider the problem 2 under Assumptions 1, 2, 3. Then, Algorithm 3 with tuning*

$$\gamma_\pi = \frac{\lambda}{4L^2}, \quad \gamma_\theta \leq \frac{1}{184\kappa^4 L}$$

*produces such $\{(\theta^t, \pi^t)\}_{t=1}^T$, that*

$$D_\psi(\pi^*(\theta^{t+1}), \pi^{t+1}) \leq \left( 1 - \frac{1}{64\kappa^2} \right) D_\psi(\pi^*(\theta^t), \pi^t) + 264\gamma_\theta^2 \kappa^6 \| \nabla \Phi(\theta^t) \|^2 + \frac{132\gamma_\theta^2 \kappa^6 \sigma^2}{B},$$

*where $\kappa = {}^L/\lambda$ is the condition number of $\mathcal{L}(\theta, \pi)$ in $\pi$.*

*Proof.* To begin, we use equation 3 in the form

$$\begin{aligned} D_\psi(\pi^*(\theta^{t+1}), \pi^{t+1}) = &D_\psi(\pi^*(\theta^{t+1}), \pi^*(\theta^t)) + D_\psi(\pi^*(\theta^t), \pi^{t+1}) \\ &+ \langle \nabla\psi(\pi^*(\theta^t)) - \nabla\psi(\pi^{t+1}), \pi^*(\theta^{t+1}) - \pi^*(\theta^t) \rangle. \end{aligned} \tag{17}$$

Further, we write the optimality condition for Line 6:

$$\left\langle -\gamma_\pi \nabla_\pi \mathcal{L}(\theta^t, \pi^t) + [\nabla\psi(\pi^{t+1}) - \nabla\psi(\pi^t)], \pi^*(\theta^t) - \pi^{t+1} \right\rangle \geq 0.$$

Applying equation 3, we obtain

$$-\gamma_\pi \left\langle \nabla_\pi \mathcal{L}(\theta^t, \pi^t), \pi^*(\theta^t) - \pi^{t+1} \right\rangle + D_\psi(\pi^*(\theta^t), \pi^t) - D_\psi(\pi^*(\theta^t), \pi^{t+1}) - D_\psi(\pi^{t+1}, \pi^t) \geq 0.$$

After re-arranging the terms, we get

$$D_\psi(\pi^*(\theta^t), \pi^{t+1}) \leq D_\psi(\pi^*(\theta^t), \pi^t) - D_\psi(\pi^{t+1}, \pi^t) - \gamma_\pi \left\langle \nabla_\pi \mathcal{L}(\theta^t, \pi^t), \pi^*(\theta^t) - \pi^{t+1} \right\rangle. \tag{18}$$

Since $\pi^*(\theta^t)$ is the exact maximum of $\mathcal{L}(\theta^t, \pi)$ in $\pi$, there is another optimality condition

$$\gamma_\pi \left\langle \nabla_\pi \mathcal{L}(\theta^t, \pi^*(\theta^t)), \pi^*(\theta^t) - \pi \right\rangle \geq 0.$$

Substituting $\pi = \pi^{t+1}$ and summing it with equation 18, we derive

$$\begin{aligned} D_\psi(\pi^*(\theta^t), \pi^{t+1}) \leq &D_\psi(\pi^*(\theta^t), \pi^t) - D_\psi(\pi^{t+1}, \pi^t) \\ &+ \gamma_\pi \left\langle \nabla_\pi \mathcal{L}(\theta^t, \pi^*(\theta^t)) - \nabla_\pi \mathcal{L}(\theta^t, \pi^t), \pi^*(\theta^t) - \pi^{t+1} \right\rangle \\ \leq &D_\psi(\pi^*(\theta^t), \pi^t) - D_\psi(\pi^{t+1}, \pi^t) \\ &+ \gamma_\pi \left\langle \nabla_\pi \mathcal{L}(\theta^t, \pi^*(\theta^t)) - \nabla_\pi \mathcal{L}(\theta^t, \pi^t), \pi^*(\theta^t) - \pi^t \right\rangle \\ &+ \gamma_\pi \left\langle \nabla_\pi \mathcal{L}(\theta^t, \pi^*(\theta^t)) - \nabla_\pi \mathcal{L}(\theta^t, \pi^t), \pi^t - \pi^{t+1} \right\rangle. \end{aligned}$$

Now, we are going to utilize the strong concavity of $\mathcal{L}(\theta, \pi)$ in $\pi$:

$$\gamma_\pi \left\langle \nabla_\pi \mathcal{L}(\theta^t, \pi^*(\theta^t)) - \nabla_\pi \mathcal{L}(\theta^t, \pi^t), \pi^*(\theta^t) - \pi^t \right\rangle \leq \frac{-\gamma_\pi \lambda}{2} D_\psi(\pi^*(\theta^t), \pi^t).$$

Thus, we have

$$D_\psi(\pi^*(\theta^t), \pi^{t+1}) \leq \left(1 - \frac{\gamma_\pi \lambda}{2}\right) D_\psi(\pi^*(\theta^t), \pi^t) - D_\psi(\pi^{t+1}, \pi^t)$$
$$+ \gamma_\pi \left\langle \nabla_\pi \mathcal{L}(\theta^t, \pi^*(\theta^t)) - \nabla_\pi \mathcal{L}(\theta^t, \pi^t), \pi^t - \pi^{t+1} \right\rangle.$$

Next, we apply Cauchy-Schwartz inequality to the scalar product and obtain

$$D_\psi(\pi^*(\theta^t), \pi^{t+1}) \leq \left(1 - \frac{\gamma_\pi \lambda}{2}\right) D_\psi(\pi^*(\theta^t), \pi^t) - D_\psi(\pi^{t+1}, \pi^t)$$
$$+ \frac{\gamma_\pi \alpha}{2} \|\nabla_\pi \mathcal{L}(\theta^t, \pi^*(\theta^t)) - \nabla_\pi \mathcal{L}(\theta^t, \pi^t)\|^2 + \frac{\gamma_\pi}{2\alpha} \|\pi^t - \pi^{t+1}\|^2.$$

Using $L$-smoothness of $\mathcal{L}$ (see Assumption 1), we obtain

$$D_\psi(\pi^*(\theta^t), \pi^{t+1}) \leq \left(1 - \frac{\gamma_\pi \lambda}{2}\right) D_\psi(\pi^*(\theta^t), \pi^t) - D_\psi(\pi^{t+1}, \pi^t)$$
$$+ \frac{\gamma_\pi \alpha L^2}{2} \|\pi^*(\theta^t) - \pi^t\|^2 + \frac{\gamma_\pi}{2\alpha} \|\pi^t - \pi^{t+1}\|^2.$$

Since $\psi$ is 1-strongly convex (see Assumption 2), we have

$$\frac{1}{2} \|\pi_1 - \pi_2\|^2 \leq D_\psi(\pi_1, \pi_2).$$

Thus,

$$D_\psi(\pi^*(\theta^t), \pi^{t+1}) \leq \left(1 - \frac{\gamma_\pi \lambda}{2}\right) D_\psi(\pi^*(\theta^t), \pi^t) - D_\psi(\pi^{t+1}, \pi^t)$$
$$+ \gamma_\pi \alpha L^2 D_\psi(\pi^*(\theta^t), \pi^t) + \frac{\gamma_\pi}{\alpha} D_\psi(\pi^t, \pi^{t+1}).$$

Choose $\alpha = \gamma_\pi$. We can derive

$$D_\psi(\pi^*(\theta^t), \pi^{t+1}) \leq \left(1 - \frac{\gamma_\pi \lambda}{2} + \gamma_\pi^2 L^2\right) D_\psi(\pi^*(\theta^t), \pi^t).$$

Since $\gamma_\pi = \lambda/4L^2$, we have

$$D_\psi(\pi^*(\theta^t), \pi^{t+1}) \leq \left(1 - \frac{1}{16\kappa^2}\right) D_\psi(\pi^*(\theta^t), \pi^t). \tag{19}$$

Let us return to equation 17. Note that

$$\nabla \psi(\pi^*(\theta^t)) - \nabla \psi(\pi^{t+1}) = \frac{1}{\lambda} \left(\nabla_\pi \mathcal{L}(\theta^t, \pi^{t+1}) - \nabla_\pi \mathcal{L}(\theta^t, \pi^*(\theta^t))\right).$$

Thus, there is

$$D_\psi(\pi^*(\theta^{t+1}), \pi^{t+1}) = D_\psi(\pi^*(\theta^{t+1}), \pi^*(\theta^t)) + D_\psi(\pi^*(\theta^t), \pi^{t+1})$$
$$+ \frac{1}{\lambda} \langle \nabla_\pi \mathcal{L}(\theta^t, \pi^{t+1}) - \nabla_\pi \mathcal{L}(\theta^t, \pi^*(\theta^t)), \pi^*(\theta^{t+1}) - \pi^*(\theta^t) \rangle$$
$$\leq D_\psi(\pi^*(\theta^{t+1}), \pi^*(\theta^t)) + D_\psi(\pi^*(\theta^t), \pi^{t+1})$$
$$+ \frac{\alpha L^2}{\lambda} D_\psi(\pi^*(\theta^t), \pi^{t+1}) + \frac{1}{\lambda \alpha} D_\psi(\pi^*(\theta^{t+1}), \pi^*(\theta^t)).$$

Let us choose $\alpha = \lambda^3/32L^4$. With such a choice, we have

$$D_\psi(\pi^*(\theta^{t+1}), \pi^{t+1}) \leq 33\kappa^4 D_\psi(\pi^*(\theta^{t+1}), \pi^*(\theta^t)) + \left(1 + \frac{1}{32\kappa^2}\right) D_\psi(\pi^*(\theta^t), \pi^{t+1}).$$

To deal with $D_\psi(\pi^*(\theta^t), \pi^{t+1})$, we utilize equation 19. As a result, we obtain

$$D_\psi(\pi^*(\theta^{t+1}), \pi^{t+1}) \leq 33\kappa^4 D_\psi(\pi^*(\theta^{t+1}), \pi^*(\theta^t)) + \left(1 - \frac{1}{32\kappa^2}\right) D_\psi(\pi^*(\theta^t), \pi^t). \tag{20}$$

The rest thing is to prove that the descent step does not dramatically change the distance between the optimal values of weights. Let us write down two optimality conditions:

$$\langle \nabla_\pi \mathcal{L}(\theta^t, \pi^*(\theta^t)), \pi - \pi^*(\theta^t) \rangle \le 0,$$
$$\langle \nabla_\pi \mathcal{L}(\theta^{t+1}, \pi^*(\theta^{t+1})), \pi - \pi^*(\theta^{t+1}) \rangle \le 0.$$

Let us substitute $\pi = \pi^*(\theta^{t+1})$ into the first inequality and $\pi = \pi^*(\theta^t)$ into the second one. When summing them up, we have

$$\langle \nabla_\pi \mathcal{L}(\theta^t, \pi^*(\theta^t)) - \nabla_\pi \mathcal{L}(\theta^{t+1}, \pi^*(\theta^{t+1})), \pi^*(\theta^{t+1}) - \pi^*(\theta^t) \rangle \le 0. \tag{21}$$

On the other hand, we can take advantage of the strong concavity of the objective (see Lemma 1) and write

$$\langle \nabla_\pi \mathcal{L}(\theta^t, \pi^*(\theta^{t+1})) - \nabla_\pi \mathcal{L}(\theta^t, \pi^*(\theta^t)), \pi^*(\theta^{t+1}) - \pi^*(\theta^t) \rangle$$
$$\le -\frac{\lambda}{2} \left[ D_\psi(\pi^*(\theta^t), \pi^*(\theta^{t+1})) + D_\psi(\pi^*(\theta^{t+1}), \pi^*(\theta^t)) \right]. \tag{22}$$

Combining equation 21 and equation 22, we obtain

$$\frac{\lambda^2}{4} \left[ D_\psi(\pi^*(\theta^t), \pi^*(\theta^{t+1})) + D_\psi(\pi^*(\theta^{t+1}), \pi^*(\theta^t)) \right]^2 \le L^2 \|\pi^*(\theta^{t+1}) - \pi^*(\theta^t)\|^2 \|\theta^{t+1} - \theta^t\|^2.$$

Re-arranging the terms and substituting Line 5, we derive

$$\left[ D_\psi(\pi^*(\theta^t), \pi^*(\theta^{t+1})) + D_\psi(\pi^*(\theta^{t+1}), \pi^*(\theta^t)) \right] \le 4\kappa^2 \|\theta^{t+1} - \theta^t\|^2$$

$$\le 4\gamma_\theta^2 \kappa^2 \left\| \frac{1}{B} \sum_{i=1}^{B} G_\theta(\theta^t, \pi^t, \xi_i^t) \right\|^2.$$

After adding and subtracting $\nabla_\theta \mathcal{L}(\theta^t, \pi^t)$, we have

$$D_\psi(\pi^*(\theta^{t+1}), \pi^*(\theta^t)) \le 4\gamma_\theta^2 \kappa^2 \left\| \nabla_\theta \mathcal{L}(\theta^t, \pi^t) \right\|^2 + 4\gamma_\theta^2 \kappa^2 \left\| \nabla_\theta \mathcal{L}(\theta^t, \pi^t) - \frac{1}{B} \sum_{i=1}^{B} G_\theta(\theta^t, \pi^t, \xi_i^t) \right\|^2.$$

Let us take an expectation and derive

$$\mathbb{E} D_\psi(\pi^*(\theta^{t+1}), \pi^*(\theta^t)) \le \mathbb{E} 8\gamma_\theta^2 \kappa^2 \|\nabla\Phi(\theta^t)\|^2 + 8\gamma_\theta^2 \kappa^2 \|\nabla_\theta \mathcal{L}(\theta^t, \pi^t) - \nabla\Phi(\theta^t)\|^2 + \frac{4\gamma_\theta^2 \kappa^2 \sigma^2}{B}$$

$$\le \mathbb{E} 8\gamma_\theta^2 \kappa^2 \|\nabla\Phi(\theta^t)\|^2 + 16\gamma_\theta^2 \kappa^2 L^2 D_\psi(\pi^*(\theta^t), \pi^t) + \frac{4\gamma_\theta^2 \kappa^2 \sigma^2}{B}.$$

Thus, equation 20 transforms into

$$\mathbb{E} D_\psi(\pi^*(\theta^{t+1}), \pi^{t+1}) \le \mathbb{E} \left( 1 - \frac{1}{32\kappa^2} + 528\gamma_\theta^2 \kappa^6 L^2 \right) D_\psi(\pi^*(\theta^t), \pi^t) + 264\gamma_\theta^2 \kappa^6 \|\nabla\Phi(\theta^t)\|^2$$
$$+ \frac{132\gamma_\theta^2 \kappa^6 \sigma^2}{B}.$$

With $\gamma_\theta \le 1/184\kappa^4 L$, we obtain

$$\mathbb{E} D_\psi(\pi^*(\theta^{t+1}), \pi^{t+1}) \le \mathbb{E} \left( 1 - \frac{1}{64\kappa^2} \right) D_\psi(\pi^*(\theta^t), \pi^t) + 264\gamma_\theta^2 \kappa^6 \|\nabla\Phi(\theta^t)\|^2 + \frac{132\gamma_\theta^2 \kappa^6 \sigma^2}{B}.$$

This completes the proof. $\qquad\square$

Now let us proceed to the convergence proof for Algorithm 3.

**Theorem 4.** *Consider the problem 2 under Assumptions 1, 2, 3. Then, Algorithm 1 with tuning*

$$\gamma_\pi = \frac{\lambda}{4L^2}, \quad \gamma_\theta \le \sqrt{\frac{43}{92 * 33792}} \frac{1}{\kappa^4 L}, \quad B = \max\left\{ 1, \frac{\kappa^{3/2}}{\varepsilon^2} \right\}$$

*requires*

$$\mathcal{O} \left( \frac{\kappa^4 L \Delta + \kappa^2 L^2 D_\psi(\pi^*(\theta^0), \pi^0) + \kappa^{3/2} \sigma^2}{\varepsilon^2} \right) \text{ iterations}$$

*to achieve an arbitrary $\varepsilon$-solution, where $\varepsilon^2 = \frac{1}{T} \sum_{t=1}^{T-1} \|\nabla\Phi(\theta^t)\|^2$, $\Delta = \Phi(\theta^0) - \Phi(\theta^*)$. $\kappa = L/\lambda$.*

*Proof.* One can note that $\Phi$ is $3\kappa L$-smooth. Indeed,

$$
\begin{aligned}
\|\nabla\Phi(\theta_1) - \nabla\Phi(\theta_2)\|^2 =& \|\nabla_\theta\mathcal{L}(\theta_1, \pi^*(\theta_1)) - \nabla_\theta\mathcal{L}(\theta_2, \pi^*(\theta_2))\|^2 \\
\leq& L^2 \left[\|\theta_1 - \theta_2\|^2 + 2D_\psi(\pi^*(\theta_1), \pi^*(\theta_2))\right] \leq L^2 \left(1 + 4\kappa^2\right)\|\theta_1 - \theta_2\|^2 \\
\leq& 9\kappa^2 L^2\|\theta_1 - \theta_2\|^2.
\end{aligned}
$$

Thus, we can write

$$
\begin{aligned}
\Phi(\theta^{t+1}) \leq& \Phi(\theta^t) + \langle\nabla\Phi(\theta^t), \theta^{t+1} - \theta^t\rangle + 3\kappa L\|\theta^{t+1} - \theta^t\|^2 \\
=& \Phi(\theta^t) - \gamma_\theta\left\langle\nabla\Phi(\theta^t), \frac{1}{B}\sum_{i=1}^B G_\theta(\theta^t, \pi^t, \xi_i^t)\right\rangle + 3\gamma_\theta^2\kappa L\left\|\frac{1}{B}\sum_{i=1}^B G_\theta(\theta^t, \pi^t, \xi_i^t)\right\|^2 \\
=& \Phi(\theta^t) - \gamma_\theta\|\nabla\Phi(\theta^t)\|^2 + \gamma_\theta\left\langle\nabla\Phi(\theta^t), \nabla\Phi(\theta^t) - \frac{1}{B}\sum_{i=1}^B G_\theta(\theta^t, \pi^t, \xi_i^t)\right\rangle \\
& + 6\gamma_\theta^2\kappa L\|\nabla_\theta\mathcal{L}(\theta^t, \pi^t)\|^2 + 6\gamma_\theta^2\kappa L\left\|\nabla_\theta\mathcal{L}(\theta^t, \pi^t) - \frac{1}{B}\sum_{i=1}^B G_\theta(\theta^t, \pi^t, \xi_i^t)\right\|^2.
\end{aligned}
$$

Consider an expectation. We have

$$
\begin{aligned}
\mathbb{E}\Phi(\theta^{t+1}) \leq& \mathbb{E}\Phi(\theta^t) - \gamma_\theta\|\nabla\Phi(\theta^t)\|^2 + \gamma_\theta\left\langle\nabla\Phi(\theta^t), \nabla\Phi(\theta^t) - \nabla_\theta\mathcal{L}(\theta^t, \pi^t)\right\rangle \\
& + 6\gamma_\theta^2\kappa L\|\nabla_\theta\mathcal{L}(\theta^t, \pi^t)\|^2 + 6\gamma_\theta^2\kappa L\sigma^2 \\
\leq& \mathbb{E}\Phi(\theta^t) - \left(\frac{\gamma_\theta}{2} - 12\gamma_\theta^2\kappa L\right)\|\nabla\Phi(\theta^t)\|^2 \\
& + \left(\frac{\gamma_\theta}{2} + 12\gamma_\theta^2\kappa L\right)\|\nabla\Phi(\theta^t) - \nabla_\theta\mathcal{L}(\theta^t, \pi^t)\|^2 + \frac{6\gamma_\theta^2\kappa L\sigma^2}{B}.
\end{aligned}
$$

Note that

$$
-\left(\frac{\gamma_\theta}{2} - 12\gamma_\theta^2\kappa L\right) \leq -\frac{43\gamma_\theta}{92}.
$$

On the other hand,

$$
\left(\frac{\gamma_\theta}{2} + 12\gamma_\theta^2\kappa L\right) \leq \gamma_\theta.
$$

Thus, we have

$$
\begin{aligned}
\mathbb{E}\Phi(\theta^{t+1}) \leq& \mathbb{E}\Phi(\theta^t) - \frac{43\gamma_\theta}{92}\|\nabla\Phi(\theta^t)\|^2 + \gamma_\theta\|\nabla\Phi(\theta^t) - \nabla_\theta\mathcal{L}(\theta^t, \pi^t)\|^2 + 6\gamma_\theta^2\kappa L\sigma^2 \\
\leq& \mathbb{E}\Phi(\theta^t) - \frac{43\gamma_\theta}{92}\|\nabla\Phi(\theta^t)\|^2 + 2\gamma_\theta L^2 D_\psi(\pi^*(\theta^t), \pi^t) + \frac{6\gamma_\theta^2\kappa L\sigma^2}{B}.
\end{aligned}
$$

Let us denote $\delta = 1 - 1/64\kappa^2$. Lemma 7 transforms into

$$
\begin{aligned}
\mathbb{E}D_\psi(\pi^*(\theta^t), \pi^t) \leq& \mathbb{E}\delta^t D_\psi(\pi^*(\theta^0), \pi^0) + 264\gamma_\theta^2\kappa^6\sum_{j=0}^{t-1}\delta^{t-1-j}\|\nabla\Phi(\theta^j)\|^2 \\
& + \sum_{j=0}^{t-1}\delta^{t-1-j}\frac{132\gamma_\theta^2\kappa^6\sigma^2}{B}.
\end{aligned}
$$

Hence,

$$
\begin{aligned}
\Phi(\theta^{t+1}) \leq& \Phi(\theta^t) - \frac{43\gamma_\theta}{92}\|\nabla\Phi(\theta^t)\|^2 + 2\gamma_\theta L^2\delta^t D_\psi(\pi^*(\theta^0), \pi^0) \\
& + 528\gamma_\theta^3\kappa^6 L^2\sum_{j=0}^{t-1}\delta^{t-1-j}\|\nabla\Phi(\theta^j)\|^2 + \frac{6\gamma_\theta^2\kappa L\sigma^2}{B} \\
& + \sum_{j=0}^{t-1}\delta^{t-1-j}\frac{264\gamma_\theta^3\kappa^6 L^2\sigma^2}{B}.
\end{aligned}
$$

Let us sum up over the iterates $t$ and obtain

$$
\begin{aligned}
\Phi(\theta^T) \leq &\Phi(\theta^0) - \frac{43\gamma_\theta}{92} \sum_{t=1}^{T-1} \|\nabla\Phi(\theta^t)\|^2 + 2\gamma_\theta L^2 \sum_{t=1}^{T-1} \delta^t D_\psi(\pi^*(\theta^0), \pi^0) \\
&+ 528\gamma_\theta^3 \kappa^6 L^2 \sum_{t=1}^{T-1} \sum_{j=0}^{t-1} \delta^{t-1-j} \|\nabla\Phi(\theta^j)\|^2 + \sum_{t=1}^{T-1} \frac{6\gamma_\theta^2 \kappa L \sigma^2}{B} \\
&+ \sum_{t=1}^{T-1} \sum_{j=0}^{t-1} \delta^{t-1-j} \frac{264\gamma_\theta^3 \kappa^6 L^2 \sigma^2}{B}.
\end{aligned}
$$

Next, we use the property of geometric progression and write

$$
\begin{aligned}
\Phi(\theta^T) \leq &\Phi(\theta^0) - \frac{43\gamma_\theta}{92} \sum_{t=1}^{T-1} \|\nabla\Phi(\theta^t)\|^2 + 128\gamma_\theta \kappa^2 L^2 D_\psi(\pi^*(\theta^0), \pi^0) \\
&+ 33792\gamma_\theta^3 \kappa^8 L^2 \sum_{t=1}^{T-1} \|\nabla\Phi(\theta^t)\|^2 + \frac{6T\gamma_\theta^2 \kappa L \sigma^2}{B} + \frac{16896T\gamma_\theta^3 \kappa^8 L^2 \sigma^2}{B}.
\end{aligned}
$$

Since $\gamma_\theta \leq \frac{1}{184\kappa^4 L}$, we can estimate this as

$$
\begin{aligned}
\Phi(\theta^T) \leq &\Phi(\theta^0) - \frac{43\gamma_\theta}{92} \sum_{t=1}^{T-1} \|\nabla\Phi(\theta^t)\|^2 + 128\gamma_\theta \kappa^2 L^2 D_\psi(\pi^*(\theta^0), \pi^0) \\
&+ 33792\gamma_\theta^3 \kappa^8 L^2 \sum_{t=1}^{T-1} \|\nabla\Phi(\theta^t)\|^2 + \frac{\gamma_\theta T \sigma^2}{B\kappa^3} + \frac{92\gamma_\theta T \sigma^2}{B}.
\end{aligned}
$$

Choosing $\gamma_\theta \leq \sqrt{\frac{43}{92*33792}} \frac{1}{\kappa^4 L}$, we derive

$$
\frac{1}{T} \sum_{t=1}^{T-1} \|\nabla\Phi(\theta^t)\|^2 \leq \mathcal{O}\left( \frac{\kappa^4 L \Delta_\Phi}{T} + \frac{\kappa^2 L^2 D_\psi(\pi^*(\theta^0), \pi^0)}{T} + \frac{\sigma^2}{B\kappa^3} + \frac{92\sigma^2}{B} \right).
$$

Let us choose $B = T/\kappa^{3/2}$ and obtain

$$
\frac{1}{T} \sum_{t=1}^{T-1} \|\nabla\Phi(\theta^t)\|^2 \leq \mathcal{O}\left( \frac{\kappa^4 L \Delta_\Phi}{T} + \frac{\kappa^2 L^2 D_\psi(\pi^*(\theta^0), \pi^0)}{T} + \frac{\kappa^{3/2}\sigma^2}{T} \right).
$$

This finishes the proof. $\qquad\square$

Note that the same reasoning could be done for the special case of a regularized simplex. Then we would obtain improved rates.

## THE USE OF LARGE LANGUAGE MODELS (LLMS)

Language models were used to improve text quality (mostly to correct grammatical errors). LLMs were not used to obtain theoretical results or write code.

