# OpenReview forum: "Enhancing Stability of Physics-Informed Neural Network Training Through Saddle-Point Reformulation"
_ICLR.cc/2026/Conference — ICLR 2026 Poster_

### Official Review · Reviewer_xUeL · 2025-10-21

**Soundness:** 3
**Presentation:** 3
**Contribution:** 2
**Rating:** 4
**Confidence:** 5

**Summary:**

This paper addresses a fundamental challenge in training Physics-Informed Neural Networks (PINNs): the instability and poor performance caused by conflicting gradients from the multiple loss terms (e.g., PDE residuals and boundary conditions). The authors propose a novel solution by reformulating PINN training as a nonconvex-strongly concave saddle-point problem (SPP). The authors provide a solid theoretical foundation for their proposed Bregman Gradient Descent Ascent (BGDA) algorithm, proving its convergence to a stationary point even in this non-Euclidean geometry setting. Extensive experiments on the comprehensive PINNacle benchmark demonstrate that AdaptiveBGDA significantly outperforms a wide range of state-of-the-art optimizers and weighting schemes across different PDE problems.

**Strengths:**

1. The saddle-point reformulation is a principled and theoretically grounded approach to the known problem of loss imbalance in PINNs..

2. The results are extensive, covering wide variety of PDEs (Poisson, Heat, Navier-Stokes, etc.) and challenging features (complex geometry, multiple domains, long time intervals).

3. The proposed AdaptiveBGDA algorithm is a concrete and practical contribution that can be readily adopted by other researchers and practitioners to improve the stability and performance of their PINN models.

**Weaknesses:**

1. Although rich experiments are conducted based on PINNacle, the relative error values in Table 3 are relatively bad and did not reflect the state of the art results, and the improvement seems modest. For example, the Poisson and Heat equations are simple and can be easily solved by traditional numerical methods with high accuracy, however, the errors stays around $\mathcal{O}(10^{-2})$, which are too large. For the complex chaotic KS equation, the improvements are from 9.57E-1 to 9.53E-1, however, for the same solution, work [a] in the year 2023 has reported a much better result 1.61E-1.
[a] AN EXPERT’S GUIDE TO TRAINING PHYSICS-INFORMED NEURAL NETWORKS, 2023.

2. The experiments in the main text seem to use a standard PINN architecture.  However, vanilla PINN architectures are hard to present the sota results[b]. While Appendix A shows results on more advanced architectures, a deeper discussion on how the saddle-point optimization interacts with specific architectural choices (e.g., Fourier feature networks, transformers) would be valuable.
[b] Causality-enhanced Discreted Physics-informed Neural Networks for Predicting
Evolutionary Equations,2024

3. While the paper shows a runtime speedup, the saddle-point formulation inherently introduces additional computational overhead per iteration due to the proximal step for updating π

4. The method introduces new hyperparameters (λ for the regularizer, step sizes γ_θ and γ_π). Although the paper uses fixed values across all benchmarks to show robustness, optimal performance in new, highly complex problems might still require tuning.

**Questions:**

1. See weakness above.

2. Could you give insights on how practitioners should go about selecting the regularization parameter λ and the initial learning rates for a new problem?

3. The experiments on high dimension is simply on d=5 dimensions, which is very weak indeed. Could you report results on higher dimensions like d=100?

4. The theory and experiments consider a small number of loss terms (M). How would the method scale and perform if M were very large, for instance, in problems with a very large number of boundary condition constraints or coupled multi-physics systems?

5. Assumption 1 needs the Lipschitz continuous for the loss function, which is widely used in the optimization literature. However, since the PINN loss is highly non-convex and ill-conditioned, does assumption 1 really hold for the PINN loss? How about the convergence

---

> ### Author Response · Authors · 2025-11-21
>
> Dear Reviewer xUeL!
>
> Thank you for your time and assessment of our work! Further, we are responding to your concerns and questions.
>
> >Poisson and Heat equations are simple and can be easily solved by traditional numerical methods with high accuracy, however, the errors stays around O(10−2)...
>
> We appreciate this insightful comment! Indeed, in our experiments, vanilla PINN exhibits lower accuracy compared to traditional methods such as FEM. This phenomenon is well-known in the literature and is observed even for advanced architectures [1, 2]. Nevertheless, PINNs are actively studied (see Table 2 in [3], including 8 A* papers). The main advantage of PINNs lies in their ability to evaluate solutions several orders of magnitude faster than non-neural methods [4,5]. Therefore, we believe that a direct comparison between PINNs and traditional numerical approaches may be incorrect.
>
> >For the complex chaotic KS equation, the improvements are from 9.57E-1 to 9.53E-1, however, for the same solution, work [a] in the year 2023 has reported a much better result 1.61E-1.
>
> We have checked [a] and its code and observed that vanilla PINN indeed achieves L2RE of $1.61E-1$ on the same chaotic KS equation. However, [a] employs $198,401$ trainable parameters (see the second paragraph in Section 7), whereas our experiments use only $40,801$ (see our anonymous repository). Given this fact, we do not see any conflict between our results and prior work. To mitigate potential misunderstandings for future readers, we have added a description of the model architecture in Section 6.
>
> >vanilla PINN architectures are hard to present the sota results[b]...
>
> We agree that our claims require validation on SOTA architectures. To address this issue, we focus on [a] from the previous comment. This work investigates modifications such as Fourier features and random weight factorization. To carry out this experiment, we integrated BGDA into the code of [a]. Among PDEs in [a], only KS matches exactly with PINNacle. We will include experiments on PINNacle in the final version of the paper. We replace LRA-like weighting with AdaptiveBGDA and perform a comparison on KS.
> |Approach|L2RE|
> |-|-|
> |LRA-like|2.58E-4|
> |AdaptiveBGDA|2.44E-4|
>
> >...saddle-point formulation inherently introduces additional computational overhead per iteration...
>
> Below, we provide a report on the iteration cost and memory usage of AdaptiveBGDA and competing methods. All runs were conducted sequentially on the Poisson 2d-C problem from PINNacle using a single NVIDIA A100 80GB.
> |Method|Time of 1000 iterations (Sec)|Peak GPU utilization (GB)|
> |-|-|-|
> |Adam|7.69|0.36|
> |AL-PINN|7.71|0.37|
> |L-BFGS|520.41|0.40|
> |LRA|20.75|0.77|
> |Multiadam|13.06|0.69|
> |NTK |18.43|0.70|
> |AdaptiveBGDA|7.64|0.37|
>
> The time complexity did not increase compared to baseline approaches due to existence of analytical expression for the ascent step when using KL divergence (see Line 252 of the original PDF). Memory requirements also did not increase significantly, as the number of weights (*no more than 11 for PINNacle PDEs*) is significantly smaller than the number of parameters (*40K*). To compute gradients over $\pi$, one needs to perform only the forward pass. We also note that LRA and NTK, which achieve SOTA results on PINNacle, underperform BGDA in terms of both time and space complexity. We have added a discussion of computational and space complexity in Section 6.4.
>
> >...optimal performance in new, highly complex problems might still require tuning.
>
> We agree that achieving optimal performance may require tuning. However, suboptimal hyperparameters do not dramatically affect the quality of AdaptiveBGDA. To confirm this numerically, we study the robustness to hyperparameters. In this experiment, we use Burgers 1D-C from PINNacle.
> |$\gamma_{\theta}$|$\gamma_{\pi}$|L2RE|
> |-|-|-|
> |0.001|0.01|1.26E-2|
> |0.001|0.1|1.30E-2|
> |0.001|0.5|1.28E-2|
> |0.004|0.01|1.30E-2|
> |0.004|0.1|1.31E-2|
> |0.004|0.5|1.31E-2|
> |0.016|0.01|1.31E-2|
> |0.016|0.1|1.30E-2|
> |0.016|0.5|1.35E-2|
>
> Next, we fix the optimal  $\gamma_{\theta}$ and $\gamma_{\pi}$ from the previous table and investigate the robustness with respect to $\lambda$.
> | $\lambda$ | L2RE |
> |-|-|
> |0.001|1.30E-2|
> |0.005 |1.26E-2|
> |0.01|1.26E-2|
> |0.05|1.31E-2|
>
> Results are consistent with our observation that, once the algorithm is tuned on Poisson 2d-C, it dominates on 72.7% of PDEs from PINNacle.
>
> For answers to the questions, we ask Reviewer to refer to our following comment.
>
> ---
>
> **References**
>
> [1] Zhao, Z., et al. “Pinnsformer: A transformer-based...”. **ICLR-24**
>
> [2] Chen, K., et al. “Pseudo-Physics-Informed Neural Operators...”. **TMLR**, 2025
>
> [3] Luo, K., et al. “Physics-informed neural networks for PDE problems...”, Springer, 2025
>
> [4] Grossmann, T., et al. “CAN PHYSICS-INFORMED NEURAL NETWORKS...”. IMA Journal of Applied Mathematics, 2024
>
> [5] Liu, X., et al. “Multi-resolution partial differential equations...”. **Nature**, 2024

---

> ### Author Response · Authors · 2025-11-21
>
> Questions:
>
> 2. Firstly, $\lambda$ controls the deviation of weights from the uniform distribution. The closer the optimal weight distribution to the uniform, the larger $\lambda$ should be. In practice, we run optimization via Adam and examine the dynamics of $L_{r,i}$ and $L_{b,j}$ (see Equation (2)) during first iterations. The initial value $\lambda^0$ is chosen as the ratio of the difference between losses at the last iteration to the sum of their absolute values. However, since the same $\lambda$ performs well across all experiments, tuning can be initialized from the value used in our paper. Secondly, the choice of $\gamma_{\pi}^0$ is based on Lemma 4, which establishes a direct proportionality of this hyperparameter to $\lambda$. Since in our experiments $\gamma_{\pi}/\lambda=10$, for a new task it makes sense to initialize $\gamma_{\pi}^0=10\lambda^0$. Finally, since the descent step in AdaptiveBGDA coincides with Adam, we recommend starting its tuning from the standard value $\gamma_{\theta}^0=0.001$.
>
> 3. Below, we present a study of L2RE, iteration cost and space complexity of AdaptiveBGDA as a function of the problem dimension for PNd from PINNacle.
> |d|L2RE|Time of 1000 iterations (Sec)|Peak GPU utilization (Gb)|
> |-|-|-|-|
> |2|5.0E-5|22.63|1.41|
> |5|1.20E-4|39.61|2.52|
> |10|4.10E-4|74.78|5.68|
> |15|5.00E-4|108.32|8.37|
>
> When d=100, one can expect: L2RE=5.98e-3, Time=764.7, GPU=44.25.
>
> 4. Our theoretical analysis answers this question. Conditions of Lemma 5 imply $L_{\pi} \sim \lambda M^2$. Substituting this into the result of Theorem 3, we conclude that BGDA scales as $M^2$.
>
> 5. We agree that the ill-conditionedness of the PINN optimization landscape renders Assumption 1 impractical. Generally, it does not hold for arbitrary neural networks [6]. However, we point out that Lipschitz continuity (Assumption 1) is commonly assumed in prior works on PINNs (see Assumption 1 in [7]; Theorem 4.5 in [8]; Assumption 3.2 in [9]; Theorem 1 in [10]). Theoretical insights from these studies are consistent with empirical observations.
>
> We also encourage Reviewer to check the revised PDF. We have made several adjustments to the structure of the experimental section to improve the overall logical coherence.
>
> If Reviewer has any remaining questions after our response, we would be happy to address them during the author-reviewer discussion. If all aspects have been clarified to Reviewer's satisfaction, we would kindly ask them to reconsider the evaluation.
>
> ---
>
> **References:**
>
> [6] Cybenko, G. “Approximation by superpositions of a sigmoidal function”. Mathematics of control, signals and systems, 1989, Springer.
>
> [7] Li, Y., et al. “Implicit Stochastic Gradient Descent for Training Physics-Informed Neural Networks”. **AAAI-23**.
>
> [8] Hwang, Y., Lim, D. “Dual Cone Gradient Descent for Training Physics-Informed Neural Networks”. **NeurIPS-24**.
>
> [9] Wu, H., et al. “RoPINN: Region Optimized Physics-Informed Neural Networks”. **NeurIPS-24**.
>
> [10] Liu, Q., et al. “CONFIG: TOWARDS CONFLICT-FREE TRAINING OF PHYSICS INFORMED NEURAL NETWORKS”. **ICLR-25**

---

> ### Comment · Reviewer_xUeL · 2025-11-24
>
> Thank you for the substantial effort the authors have invested in this work. However, I still feel that the improvements demonstrated are relatively narrow for a top conference. In Table 3, the best and second-best results are extremely close (e.g., 1.30E-2(BGDA) vs 1.33E-2(LBFGS), 7.78E-1(BGDA) vs 1.09E+0(MultiAdam)). Such small gaps may stem from randomness or hyperparameter tuning, especially since no error bars are reported. In addition, the errors of the latter group are quite large, making the comparison less informative.
>
> Given that the PINN community has developed rapidly over the past six years, I believe that the modest improvements shown here may not be sufficient for acceptance at ICLR. Therefore, I will maintain my current score.

---

> > ### Author Response · Authors · 2025-11-28
> >
> > Thanks to Reviewer for participating in the discussion!
> >
> > >small gaps may stem from randomness or hyperparameter tuning, especially since no error bars are reported
> >
> > To address Reviewer’s concern regarding randomness, we have included deviations computed over 5 runs for all experiments in Table 3. Below, we provide a comparison for PDEs mentioned by Reviewer.
> > ||Adam|LBFGS|LRA|NTK|RAR|MultiAdam|BGDA|
> > |-|-|-|-|-|-|-|-|
> > |Burgers 1d-C|(1.44$\pm$0.04)E-2|(1.33$\pm$0.01)E-2|(2.66$\pm$0.33)E-2|(1.90$\pm$0.02)E-2|(3.10$\pm$0.02)E-2|(4.96$\pm$0.38)E-2|(1.29$\pm$0.01)E-2|
> > |Wave 2d-CG|(1.66$\pm$0.02)E+0|(1.33$\pm$0.00)E+0|(1.53$\pm$0.10)E+0|(2.09$\pm$0.15)E+0|(1.21$\pm$0.09)E+0|(1.08$\pm$0.02)E+0|(7.80$\pm$0.03)E-1|
> >
> > It can be seen that intervals do not overlap, indicating that BGDA outperforms baselines with statistical significance.
> >
> > >In Table 3, the best and second-best results are extremely close
> >
> > In this comment, Reviewer focuses on a few rows from Table 3. However, in **18\%** of its rows, we outperform the closest competitor by more than a factor of two; in **36\%** of its rows, BGDA improves L2RE by more than **30\%**; and in **77.2\%** of all cases, the proposed method dominates the nearest baseline.
> >
> > At the same time, the previous best optimizer dominated only 27.3\% of the benchmark (**2.83 times worse** than our result), and among those cases, only one out of 22 PDEs shows an improvement larger than 5.4%. Therefore, we believe that our work brings a substantial improvement to the landscape of optimizers for PINNs.
> >
> > >modest improvements shown here may not be sufficient for acceptance
> >
> > In the domain of optimizers for PINNs, even modest improvements on individual PDEs are considered significant. We would like to draw the Reviewer’s attention to several related works.
> >
> > [11] (**ICML-23**) highlights substantial gains on a few PDEs as a key strength of their approach. However, without considering BGDA, the proposed method shows improvements exceeding 12% only on a small fraction of PDEs, while underperforming on 86% of PINNacle.
> >
> > [12] (**ICLR-25 spotlight**) presents a method that provides noticeable improvements only on Schrödinger, while showing almost no advantage over existing optimizers on Burgers and performing worse on Kovasznay and Beltrami (see Figure 4).
> >
> > Moreover, in some areas of deep learning, even small improvements in performance are considered meaningful. Let us consider the well-studied task of machine learning on tabular data as an example. [13] (**ICLR-24**) reports an improvement of around 1% over a standard MLP baseline (see Table 2). This was further increased by 1% in [14] (**ICLR-25**). At the same time, [15] (**NeurIPS-24**) demonstrates only modest gains of advanced architectures compared to even boosted trees.
> >
> > We believe that our responses, along with new experiments, have fully addressed the concerns Reviewer raised. We hope they will consider these improvements and provide updated feedback.
> >
> > ---
> >
> > **References:**
> >
> > [11] Yao, J., et al. “MultiAdam: Parameter-wise…”. **ICML-23**
> >
> > [12] Liu, Q., et al. “CONFIG: TOWARDS CONFLICT-FREE...”. **ICLR-25**
> >
> > [13] Gorishniy, Y., et al. “TABR: TABULAR DEEP LEARNING…”. **ICLR-24**
> >
> > [14] Gorishniy, Y., et al. “TABM: ADVANCING TABULAR DEEP LEARNING…”. **ICLR-25**
> >
> > [15] Holzmueller, D., et al. “Better by Default: Strong Pre-Tuned MLPs…”. **NeurIPS-24**

---

### Official Review · Reviewer_akXc · 2025-10-21

**Soundness:** 3
**Presentation:** 2
**Contribution:** 2
**Rating:** 4
**Confidence:** 4

**Summary:**

Although PINNs have attracted significant attention and found wide applications, their performance often suffers from instability due to the complexity of their loss functions. To address this, the research team reformulated PINN training as a nonconvex–strongly concave saddle-point problem. They first established a solid theoretical foundation by studying this class of saddle-point problems under non-Euclidean geometry and proposing a Bregman proximal mapping–based method with rigorous guarantees on optimization dynamics. Extensive empirical validation followed, demonstrating that the proposed method consistently outperforms existing approaches and achieves state-of-the-art results across nearly all benchmark PDEs, effectively unifying previously task-specific dominant methods. Finally, the paper presents a comprehensive theoretical and empirical analysis of the proposed approach and highlights its key contributions: (1) establishing the theoretical foundation, and (2) conducting extensive empirical verification.

**Strengths:**

1.  Reformulating PINN training as a nonconvex–strongly concave saddle-point problem and introducing the AdaptiveBGDA optimizer based on Bregman divergence is innovative. The method offers a fresh perspective distinct from traditional gradient-based techniques.

2. Theoretical analyses are rigorous, with proven convergence guarantees. Experimental results across multiple PDE benchmarks show consistent improvements over state-of-the-art methods.

3. The paper is well-structured, with clear explanations of the algorithms and mathematical derivations, intuitive figures, and transparent proofs.

4. To some extent, it addresses the instability issues in PINN training and has potential application value for solving PDEs in scientific computing.

**Weaknesses:**

1. The theoretical analysis (Theorem 1 and Lemma 2) establishes convergence under idealized assumptions, but these are not explicitly connected to observed empirical behaviors. The discussion lacks explanation of how theoretical stability guarantees translate to improved convergence in practice.

→ Improvement: Add a section explicitly mapping theoretical assumptions (e.g., strong concavity, bounded divergence) to empirical observations in experiments.

2. Although the paper includes a link to anonymous code for review, it does not ensure full reproducibility—some implementation details and hyperparameter settings are missing.

→ Improvement: Provide complete training scripts, configuration files, and datasets used for each benchmark.

3. The theoretical role of Bregman divergence is mathematically well defined, but its intuitive contribution to training stability in non-Euclidean spaces is underexplained.

→ Improvement: Include visual or conceptual explanations illustrating how Bregman divergence shapes the optimization trajectory or balances competing losses.

**Questions:**

1. What is the computational complexity (in both time and space) of AdaptiveBGDA compared to Adam, NTK, or AL-PINN? Please provide either a theoretical complexity analysis or detailed runtime profiling (e.g., per iteration cost, GPU utilization, or convergence rate per epoch). This would clarify whether AdaptiveBGDA’s superior stability comes at a computational trade-off.

2. The paper mentions fixed hyperparameters (γπ = 0.1, γθ = 0.008, α1 = 0.9, α2 = 0.999, β = 0.999). How sensitive is the method’s performance to these choices? Please include a sensitivity analysis or guidelines for hyperparameter selection would strengthen the practical usability of the method.

---

> ### Author Response · Authors · 2025-11-17
>
> Dear Reviewer akXc!
>
> Thank you for comments regarding our work! Further, we are addressing your concerns.
>
> >The theoretical analysis establishes convergence under idealized assumptions ... not explicitly connected to observed empirical behaviors
>
> We agree that theoretical section of our work employs an idealized setting. However, our analysis remains within commonly accepted assumptions. Gradient Lipschitz continuity (Assumption 1) is common in prior works on PINNs (see Assumption 1 in [1]; Theorem 4.5 in [2]; Assumption 3.2 in [3]; Theorem 1 in [4]). Uniformly bounded variance (Assumption 3) is introduced solely for the analysis of StochasticBGDA and is also standard (see Assumption 2 in [5]). Despite Assumptions 1, 3 are usually not satisfied for NNs [6, 7], results obtained in the aforementioned papers are consistent with empirical observations. We also note that strong convexity of the distance-generating function (Assumption 2) is not restrictive, since commonly used distances are generated by 1-strongly convex functions.
>
> >Add a section explicitly mapping ... assumptions to empirical observations...
>
> We have added a discussion in Section 4. We thank you for the comment!
>
> >Provide complete training scripts, configuration files, and datasets used for each benchmark.
>
> The training script, as well as descriptions of hyperparameters and datasets, are available in our repository. The configuration file is not provided because we do not use one; all necessary parameters are passed directly via the command line that can be found in README.md. The benchmark description is located in README_by_pinnacle_authors.md.
>
> >Include visual or conceptual explanations illustrating how Bregman divergence ... balances competing losses.
>
> We explain the role of Bregman divergence in Section 1. Let us describe it in more detail. Consider $S$ to be unit simplex. Then the maximization step searches for an optimal discrete distribution conditioned on model parameters. Euclidean distance is inadequate in this case. For example, if some weight is zero in one distribution but nonzero in another, the distance between them should be infinite, since one of them completely rules out an outcome of another one. The Euclidean metric, however, fails to capture this behavior. For conceptual explanation, see Figure 1.1 in [8]. To support our claim, we provide a comparison with a competing Euclidean dual-dimer method (we mention this algorithm in Line 77).
> |Problem|Dual-dimer|AdaptiveBGDA|
> |-|-|-|
> |Poisson 2d-CG|7.26E-2|1.76E-2|
> |Poisson 3d-CG|1.57E-1|4.78E-2|
> |Wave 1d-C|2.64E-1|1.62E-2|
> |High dim PNd|4.2E-4|1.20E-4|
> |Inverse HInv|1.08E+0|4.05E-2|
>
> See Appendix B for the full set of results and discussion.
>
> **Questions:**
>
> 1) We provide a report on time/space complexity of BGDA and competing methods. We are ready to consider other problems from PINNacle if necessary.
> |Method|Time of 1000 iterations (Sec)|Peak GPU utilization (GB)|
> |-|-|-|
> |Adam|7.69|0.36|
> |AL-PINN|7.71|0.37|
> |L-BFGS |7.58|0.37|
> |LRA|20.75|0.77|
> |Multiadam|13.06|0.69|
> |NTK|18.43|0.70|
> |AdaptiveBGDA|7.64|0.37|
>
> The time complexity is comparable with baseline approaches due to existence of closed-form solution for the ascent step (see Line 245 of the revised PDF). Space complexity also did not increase, as the number of weights (*no more than 11 for PINNacle*) is significantly smaller than the number of parameters (*40K*). See Section 6.4 of the revised PDF for the complete discussion.
>
> 2) We examine the effect of varying step sizes on Burgers 1d-C.
> |$\gamma_{\theta}$|$\gamma_{\pi}$|L2RE|
> |-|-|-|
> |0.001|0.01|1.26E-2|
> |0.001|0.1|1.30E-2|
> |0.001|0.5|1.28E-2|
> |0.004|0.01|1.30E-2|
> |0.004|0.1|1.31E-2|
> |0.004|0.5|1.31E-2|
> |0.016|0.01|1.31E-2|
> |0.016|0.1|1.30E-2|
> |0.016|0.5|1.35E-2|
>
> We are ready to provide a similar study for other problems from PINNacle if necessary. We have added this results to Appendix C of the revised PDF.
>
> We also have addressed certain inaccuracies in the text and restructured the experimental section for clarity in the revised PDF.
>
> If you have any remaining questions, we would be happy to address them during the discussion period. If all aspects have been clarified to your satisfaction, we would kindly ask you to reconsider the evaluation.
>
> ---
>
> **References:**
>
> [1] Li, Y., et al. “Implicit Stochastic Gradient Descent for...”. **AAAI-23**
>
> [2] Hwang, Y., Lim, D. “Dual Cone Gradient Descent for....”. **NeurIPS-24**
>
> [3] Wu, H., et al. “RoPINN: Region Optimized...”. **NeurIPS-24**
>
> [4] Liu, Q., et al. “CONFIG: TOWARDS CONFLICT-FREE...”. **ICLR-25**
>
> [5] Huang, F., et al. “Efficient Mirror Descent Ascent Methods for...”. **NeurIPS-21**
>
> [6] Cybenko, G. “Approximation by superpositions...”. Mathematics of control, signals and systems, 1989, Springer
>
> [7] Patel, V. “Counterexamples for Noise Models of Stochastic Gradients”. Examples and Counterexamples, 2023, Elsevier
>
> [8] Dragomir, R. “Bregman gradient methods for relatively-smooth optimization”. PhD Thesis

---

> > ### Comment · Reviewer_akXc · 2025-11-21
> >
> > Thanks for the timely and detailed response the authors have made. Most of my concerns have been addressed. A few remaining and comments for
> >
> > W1. I simply checked the Lipschitz assumption in [1], however, it is the Lipschitz assumption for activation functions, which holds true for tanh,sin,etc commonly used activations in PINNs community. However, in the authors' paper, it is for the loss function. I didn't check others. By the way, what's the role of  Theorem 1 in the paper for authors using these assumptions to proof? Can you numerically test to match Theorem 1 (for example, epsilon/2 accuracy needs 4 times iterations)
> >
> > Q1. I appreciate the report on the numerical complexity of BGDA and competing methods, and Figure 4 in the paper. Some doubts: Why AdaptiveBGDA(7.64) even smaller than Adam(7.69)? How is the L-BFGS(7.58) computed since it is a second order methods with lots of matrix computed? Can you give a theoretical complexity comparison if possible? It is also ok if not provided since the numerical comparison is enough. For Figure 4, a suggestion is to plot the training loss decay for different optimizers.

---

> > > ### Author Response · Authors · 2025-11-24
> > >
> > > We are glad that we have addressed most of Reviewer’s concerns! Below, we respond to the remaining ones.
> > >
> > > > I simply checked the Lipschitz assumption in [1], however, it is the Lipschitz assumption for activation functions, which holds true for tanh,sin,etc commonly used activations in PINNs community. However, in the authors' paper, it is for the loss function.
> > >
> > > Typically for works on PINNs, [1] employs an MSE loss. Let us consider $$L(x)=\frac{1}{2}||u_{true}-u(x)||^2.$$ Then
> > >  $$\nabla^2 L(x)=\nabla u(x)\nabla u(x)^{\top}-(u_{true}-u(x))\nabla^2u(x).$$ Since authors of [1] assume activations and their gradients to be Lipschitz-continuous, we have $||\nabla u(x)||\leq C_1$ and $||\nabla^2 u(x)||\leq C_2$. Moreover, there exists $E$ that bounds $|u_{true}-u(x)|$, as the optimization process converges in a finite number of iterations. Therefore, $$||\nabla^2L(x)||\leq C_1^2+EC_2,$$ which implies Lipschitz continuity of the objective $L(x)$. We also refer Reviewer to [2], [3], [4], cited in our rebuttal, where the Lipschitz continuity of the objective is introduced directly.
> > >
> > > We also point out that, without Lipschitz continuity, it is impossible to prove the convergence of any gradient-based method (see Section 4 in [9] for an example where gradient descent produces objective function values that diverge). Thus, the assumption of gradient Lipschitz continuity is required for any convergence theory in DL, not just for PINNs. Nevertheless, empirical observations show that losses of neural networks are reasonably well described by this assumption (see the study of Lipschitz constant dynamics in [10]).
> > >
> > > >what's the role of Theorem 1 in the paper for authors using these assumptions to proof?
> > >
> > > Theorem 1 provides worst-case convergence guarantees for BGDA. There is no PDE instance for which training a PINN would exhibit a worse upper complexity bound than the one established in our work.
> > >
> > > >Can you numerically test to match Theorem 1 (for example, epsilon/2 accuracy needs 4 times iterations)
> > >
> > > We have added this experiment and its discussion in Appendix D.
> > >
> > > >Why AdaptiveBGDA(7.64) even smaller than Adam(7.69)?
> > >
> > > Results presented in Table 4 were obtained from a single run, as we believe that it is sufficient to consider orders of magnitude rather than exact values for comparing the computational overhead. A difference of 0.05 falls within noise, which can be attributed to various factors, such as GPU temperature. We report the average cost per iteration of Adam and AdaptiveBGDA along with the standard deviation, computed over 1000 realizations.
> > > |Method|Mean|Std|
> > > |-|-|-|
> > > |Adam|0.0082|0.0021|
> > > |BGDA|0.0085|0.0024
> > >
> > > Thus, on average, Adam is slightly faster than AdaptiveBGDA. This is explained by the fact that BGDA performs additional computations during the weight update.
> > >
> > > >How is the L-BFGS(7.58) computed since it is a second order methods with lots of matrix computed?
> > >
> > > Since LBFGS indeed requires a significant amount of time, it is common practice to perform several thousand Adam iterations (5K for PINNacle) before LBFGS when training a PINN. As we measured the time over 1000 iterations, the result reported in Table 4 actually matches Adam. In the revised PDF, we have corrected this by measuring the time for a thousand LBFGS iterations. We thank Reviewer for this valuable observation!
> > >
> > > >Can you give a theoretical complexity comparison if possible? It is also ok if not provided since the numerical comparison is enough.
> > >
> > > Theoretical analysis is one of the strengths of our work. Since most of the methods in Table 4 (LRA, NTK, RAR, MultiAdam) do not have theoretically justified rates, we cannot provide a comparison of theoretical complexity. Furthermore, we could not find established rates among the direct competitors of our approach (dual-dimer, AL-PINN, see Section B).
> > >
> > > > For Figure 4, a suggestion is to plot the training loss decay for different optimizers.
> > >
> > > We have included Adam, LRA and MultiAdam in Figure 4. Thanks to Reviewer for the suggestion!
> > >
> > > ---
> > >
> > > **References:**
> > >
> > > [9] Patel, V., Berahas, A. “Gradient Descent in the Absence of Global Lipschitz Continuity of the Gradients”. **SIAM**, 2024
> > >
> > > [10] Khromov, G., Singh, S. “SOME FUNDAMENTAL ASPECTS ABOUTLIPSCHITZ CONTINUITY OF NEURAL NETWORKS”. **ICLR-24**

---

> ### Author Response · Authors · 2025-11-27
>
> Dear Reviewer akXc!
>
> We understand the demands on your time during the author–reviewer discussion and greatly appreciate the effort you devote to evaluating submissions. However, we would like to kindly follow up on our latest comment, where we addressed your two remaining concerns. Given the borderline scores for our paper, we would be grateful if you could review our responses and consider revising your evaluation, or discuss any remaining issues. Your feedback is highly valued!
>
> We sincerely appreciate your time and thoughtful consideration!

---

> > ### Comment · Reviewer_akXc · 2025-11-28
> >
> > Thanks for the reply. I mean the strong Lipschitz condition holds for PINN problems? Appendix D should be reported in detail for the experimental details, and how you plot the theoretical decay curve? Figure 4 gives the comparison for L2 error in the training process, I mean to plot the training loss (or loss gradient to reflect Theorem 1) vs iterations for different optimizers, which are more consistent with your Theorem 1.

---

> > > ### Author Response · Authors · 2025-12-01
> > >
> > > >I mean the strong Lipschitz condition holds for PINN problems?
> > >
> > > Generally, Lipschitz condition does not hold for PINN problems, as well as for other DL tasks. However, it is impossible to prove any convergence theory without this assumption (see Section 4 in [9] for an example where gradient descent produces objective function values that diverge). We have answered this question in detail in our previous comment.
> > >
> > > >Appendix D should be reported in detail for the experimental details
> > >
> > > We have included a detailed description in Appendix D. In particular, we report the procedure for constructing the theoretical learning curve and the selected constants.
> > >
> > > >how you plot the theoretical decay curve?
> > >
> > > Since the convergence bound contains constants that cannot be computed in practice ($L$, $\kappa$, $\Phi(\theta^0)-\Phi(\theta^*)$, $D_{\psi}(\pi^(\theta^0),\pi^0)$), we define the theoretical convergence function as $f(t)=\frac{C}{\sqrt{t}}$ and check whether there exists a constant $C$ such that the plot of $f(t)$ lies within the confidence interval of the curve corresponding to the gradient norm.
> > >
> > > > I mean to plot the training loss (or loss gradient to reflect Theorem 1) vs iterations for different optimizers
> > >
> > > We have plotted the loss as a function of iterations count in Appendix D. Thank you for the suggestion!
> > >
> > > ---
> > >
> > > **References:**
> > >
> > > [9] Patel, V., Berahas, A. “Gradient Descent in the Absence of Global Lipschitz Continuity of the Gradients”. SIAM, 2024

---

### Official Review · Reviewer_Me5S · 2025-10-31

**Soundness:** 2
**Presentation:** 2
**Contribution:** 2
**Rating:** 4
**Confidence:** 4

**Summary:**

The paper tackles the instability problem of PINNs by reformulating the training process as a saddle-point problem (SPP). Specifically, the authors introduce trainable loss weights $\pi$ and impose a divergence regularization term to ensure strong concavity with respect to $\pi$. This transformation is claimed to improve regularity and stability during optimization. The resulting formulation is optimized using a BGDA algorithm and its adaptive variant (AdaptiveBGDA). Theoretical guarantees are provided for convergence in nonconvex–strongly concave settings, and experimental results on multiple PDE benchmarks demonstrate faster convergence and improved accuracy compared to baseline PINN training schemes.

**Strengths:**

1. **clear motivation**: The work addresses a well-known bottleneck in PINN training: gradient imbalance between boundary and residual terms. The motivation for stabilizing the training through adaptive loss weighting is well articulated and relevant.

2. **Novel regularization for strong concavity**: Introducing a divergence-based regularization (via Bregman divergence) ensures strong concavity in the inner maximization problem, which theoretically stabilizes the optimization dynamics.

3. **simple algorithm**: The BGDA framework is conceptually simple and can be implemented with minor modifications to existing PINN optimizers.

**Weaknesses:**

1. **Overall poor presentation including unlear definitions**: for example, in Equation (1), quantities such as $M_r$, $M$, and the role of $S$ and $\hat pi$ are not clearly defined.

2. **Potentially incorrect criticism of prior work**: I think the claim that Liu & Wnag (2021) used $\pi \in \mathbb R^d$ and suffered instability due to the unconstrained weight space may be inaccurate. The dimension of $\pi$ in Liu & Wang (2021) is also the number of losses.

3. **Justification for reformulation**: Transforming a nonconvex minimization problem into a nonconvex–strongly concave SPP can make optimization harder, as it introduces an inner maximization step that must be approximated in practice. The paper lacks a quantitative discussion on whether the stability gain outweighs the computational burden..

4. **Limited theoretical scope**: The analysis guarantees stationarity of the envelope function $\Phi(\theta)=L(\theta, \pi^*(\theta))$ but does not ensure convergence to a joint stationary point. Therefore, the derived resutls are somewhat weak compared to standard convergence analysis in the literature on minimax optimization .

5. **Lack of fair computational comparison**: computational costs are not reported. Since SPP training increases per iteration cost, it should be reported.

6. **Insufficient experimental details and reproducibility**: Standard deviations are missing. Deatils for selection for hyperparameters are not rerported.

7. **Missing baseline**: Comparison with dual-dimer approaches (Liu & Wang (2021)) is missing.

**Questions:**

1. How sensitive is the algorithm to the choise of  hyperparameters?

2. Please include computational overhead of the proposed method.

3. Pleae include experimental comparison with dual dimer method

4. How does the method perform on high-dimensional or more challenging PDEs?

---

> ### Author Response · Authors · 2025-11-17
>
> Dear Reviewer Me5S!
>
> Thank you for your comments and questions! Below we are addressing your concerns.
>
> >quantities such as Mr, M, and the role of S and p^i are not clearly defined
>
> Thank you for important comment! We formulate the problem of interest as a system of $M$ equations, where indices $[1,M_r]$ and $[M_r+1,M]$ correspond to differential equations and boundary/initial conditions, respectively. $\pi^i$ is the non-negative importance of $i$-th operator. $S$ denotes the set of feasible weights. We have added clarifications in Section 1.
>
> >claim that Liu & Wnag (2021) ... may be inaccurate.
>
> We agree that Liu & Wang suggested weighting from the unit simplex rather than from $R^M$. Main issue of their approach is the use of Euclidean updates, which is supported with comparison of BGDA and dual-dimer on PINNacle.
> |Problem|Dual-dimer|AdaptiveBGDA|
> |-|-|-|
> |Poisson 2d-CG|7.26E-2|1.76E-2|
> |Poisson 3d-CG|1.57E-1|4.78E-2|
> |Wave 1d-C|2.64E-1|1.62E-2|
> |High dim PNd|4.2E-4|1.20E-4|
> |Inverse HInv|1.08E+0|4.05E-2|
>
> See Appendix B for the full set of experiments.
>
> >SPP can make optimization harder.
>
> We compare time/space complexity of AdaptiveBGDA and competing methods.
> |Method|Time of 1000 iterations (Sec)|Peak GPU utilization (GB)|
> |-|-|-|
> |Adam|7.69|0.36|
> |AL-PINN|7.71|0.37|
> |L-BFGS|520.41|0.40|
> |LRA|20.75|0.77|
> |Multiadam|13.06|0.69|
> |NTK|18.43|0.70|
> |AdaptiveBGDA|7.64|0.37|
>
> Time complexity is comparable to baseline approaches due to the existence of analytical expression for the ascent step when using KL divergence. GPU usage also do not increase, as the number of weights (*no more than 11 for PINNacle PDEs*) is much smaller than the number of parameters (*40K*). See Section 6.4 for the complete discussion.
>
> >...guarantees stationarity of the envelope function
>
> In our work, we study N-SC SPPs. This class of problems involves envelope function as a common stopping criterion (see Definition 2.4 in [1]; Introduction in [2]; Related Work in [3]), which implies convergence to a stationary point in the classic sense (see Proposition 4.12 in [4]). Our analysis follows this established approach. We have added this clarification to Section 4.3.
>
> >computational costs are not reported
>
> See the above ablation study.
>
> >Standard deviations are missing.
>
> We have included deviations in Table 3. We are willing to include deviations to all tables in the final version of our work.
>
> >selection for hyperparameters are not rerported.
>
> Let us formulate the procedure of selecting initial hyperparameters for further tuning. Firstly, $\lambda$ controls the deviation of weights from the uniform distribution. We run optimization via Adam and examine the dynamics of $L_{r,i}$ and $L_{b,j}$ (see Equation (2)) during several iterations. The initial value $\lambda^0$ is chosen as the ratio of the difference between losses at the last iteration to the sum of their absolute values. Since the same $\lambda$ performs well across all experiments, tuning can be initialized from the value suggested in our paper. Secondly, the choice of $\gamma_{\pi}^0$ is based on Lemma 4, which establishes a direct proportionality with $\lambda$. Since in our experiments $\gamma_{\pi}/\lambda=10$, for a new task one can initialize $\gamma_{\pi}^0=10\lambda^0$. Finally, since the descent step in AdaptiveBGDA coincides with Adam, we recommend starting its tuning from the standard $\gamma_{\theta}^0=0.001$.
>
> **Questions:**
>
> 1) We examine the sensitivity of AdaptiveBGDA to the choice of step sizes. We use Burgers 1d-C.
> |$\gamma_{\theta}$ |$\gamma_{\pi}$|L2RE|
> |-|-|-|
> |0.001|0.01|1.26E-2|
> |0.001|0.1|1.30E-2|
> |0.001|0.5|1.28E-2|
> |0.004|0.01|1.30E-2|
> |0.004|0.1|1.31E-2|
> |0.004|0.5|1.31E-2|
> |0.016|0.01|1.31E-2|
> |0.016|0.1|1.30E-2|
> |0.016|0.5|1.35E-2|
>
> Next, we fix optimal  $\gamma_{\theta}$ and $\gamma_{\pi}$ and vary $\lambda$.
>
> |$\lambda$|L2RE|
> |-|-|
> |0.001|1.30E-2|
> |0.005|1.26E-2|
> |0.01|1.26E-2|
> |0.05|1.31E-2|
>
> We are ready to provide a similar study for other PDEss from PINNacle if necessary. Discussion of this Table can be found in Appendix C.
>
> 2) See above.
>
> 3) See above.
>
> 4) PINNacle includes challenging problems, including high dimensional ones. Please, specify problems that we should cover in the experimental section.
>
> We also kindly ask you to check the revised PDF. We have corrected several inaccuracies in the manuscript and reorganized the experimental section.
>
> We remain available for any further discussion, and would be grateful if you could consider raising the score should you find our responses satisfactory.
>
> ---
>
> **References**
>
> [1] Zhang, S., et al. “The Complexity of Nonconvex-Strongly-Concave...”. **UAI-21**
>
> [2] Wang, N., et al. “Efficient First Order Method for...”. Journal on Optimization, 2024, SIAM
>
> [3] Xu, Q., et al. “A Stochastic GDA Method With Backtracking...”. arXiv, 2024
>
> [4] Lin, W., et al. “On gradient descent ascent for...”, **ICML 2020**

---

> ### Author Response · Authors · 2025-11-27
>
> Dear Reviewer Me5S!
>
> We would like to gently follow up on our earlier rebuttal. We understand the considerable workload during the review process and sincerely appreciate the time you devote to evaluating submissions.
>
> We kindly ask you to give feedback on our response. We would also be grateful if you could take a moment to look at new experimental results included in the revised PDF. Any comments you might have on these points would be greatly appreciated.
>
> Thank you very much for your time and consideration!

---

> > ### Comment · Reviewer_Me5S · 2025-11-27
> >
> > Thank you for the detailed rebuttal. Most of my concerns were addressed, and the revision clarified key points. I am satisfied with the response and increase my score to 6.

---

> > > ### Author Response · Authors · 2025-11-28
> > >
> > > Thank you very much for the positive evaluation of our work!
> > >
> > > We kindly remind you to **edit** the review to reflect the new score. We appreciate your time and attention!

---

### Official Review · Reviewer_Zu1m · 2025-10-31

**Soundness:** 3
**Presentation:** 4
**Contribution:** 3
**Rating:** 6
**Confidence:** 3

**Summary:**

This paper proposes to reformulate the PINN loss as a min–max (saddle-point) problem with a Bregman regularizer to stabilize the weights. Using a simple optimizer, Bregman Gradient Descent–Ascent (BGDA), the authors empirically show improved performance on numerous PDE benchmarks. To explain where the gains come from, the paper presents a gradient conflict analyses showing reduced conflict, which supports the reported performance improvements.

**Strengths:**

1. High novelty. The paper reformulates PINN into a new optimization framework that mitigates gradient-conflict issues.

2. Clear and rigorous. The problem setup and method are explained clearly, and the paper provides rich theory, such as complexity bounds, that demonstrates the strength of the approach.

3. Strong experiments. The evaluation across 22 PDEs is comprehensive and shows the stability of the proposed method.

**Weaknesses:**

Although this paper have already compared to many baseline optimizers. It would better to include some SOTA method like [1] and [2], especially [2] also experimented on Burgers equation and achieves a much lower error.


[1] Rathore, Pratik, et al. "Challenges in training pinns: A loss landscape perspective." arXiv preprint arXiv:2402.01868 (2024).

[2] Kiyani, Elham, et al. "Which Optimizer Works Best for Physics-Informed Neural Networks and Kolmogorov-Arnold Networks?." arXiv preprint arXiv:2501.16371 (2025).

**Questions:**

None.

**Details Of Ethics Concerns:**

None.

---

> ### Author Response · Authors · 2025-11-19
>
> Dear Reviewer Zu1m!
>
> Thank you for positive evaluation of our work and for suggestions on further improvements!
>
> Indeed, additional baselines would strengthen the experimental section of our work. We checked the suggested papers and observed that in [1] a small model (30 times smaller than the one used in PINNacle) achieves an anomalously low L2RE in single precision (3.60e-5 vs. 1.30e-2). We attribute this to the fact that [1] actually uses a significantly larger training dataset than PINNacle. This claim cannot be verified directly, as the authors do not specify the training grid resolution and do not provide code. However, evidence supporting our hypothesis is the peak GPU memory usage reported for training their model with 1341 trainable parameters: 24.56 GB on NVIDIA A6000, whereas we obtained only 0.36 GB on NVIDIA A100 when using a model of the same size. Therefore, it is not appropriate to compare the results in Table 1 of [2] with those in Table 3 of our work.
>
> Below, we provide a comparison of AdaptiveBGDA with SSBroyden+Wolfe [2] and NNCG [3] on Burgers 1d-C from PINNacle using MLP with 16897 trained parameters. The recommended algorithmic settings are used whenever available.
> |Method|L2RE|Time of 1000 iterations (Sec)|Peak GPU utilization (GB)|
> |-|-|-|-|
> |Adam+SSBroyden+Wolfe|1.32e-2|176.69|10.66|
> |Adam+LBFGS+NNCG|1.33e-2|13937.61|2.68|
> |AdaptiveBGDA|1.52e-2|8.25|0.24|
>
> Thus, despite the modest improvement in L2RE, the competing methods suggested by Reviewer exhibit a pronounced computational bottleneck.
>
> We also would like to point out that, in addition to outperforming optimizers implemented in PINNacle, we surpass the closest conceptual competitors (see Appendix B). In the first of these methods, PINN training is formulated as a saddle-point problem via an augmented Lagrangian (AL-PINN, see [2]); in the second, trainable weights are introduced within the unit simplex, but without accounting for the non-Euclidean geometry of $S$ (dual-dimer, see [3]). We also encourage Reviewer to check our responses to the other reviews for additional empirical results.
>
> We are ready to continue the discussion if Reviewer has any remaining concerns. If we have addressed all questions satisfactorily, we would kindly ask them to raise the score.
>
> ---
>
> **References**
>
> [1] Kiyani, Elham, et al. "Which Optimizer Works Best for Physics-Informed Neural Networks and Kolmogorov-Arnold Networks?." arXiv preprint arXiv:2501.16371 (2025).
>
> [2] Son, H., et al. “Enhanced physics-informed neural networks with augmented lagrangian relaxation method (al-pinns)”. Neurocomputing, 2023, Elsevier
>
> [3] Liu, D., Wang, Y. “A dual-dimer method for training physics-constrained neural networks with minimax architecture”, Neural Networks, 2021, Elsevier

---

> > ### Comment · Reviewer_Zu1m · 2025-11-25
> >
> > Thank you for your response and effort on new experimentation. Most of my concerns have been resolved. I recommend that the authors include the additional experiments in the revised paper to show the significant superiority in computing resources saving.

---

> ### Author Response · Authors · 2025-11-26
>
> Thanks to Reviewer for their response to our rebuttal. We have added a comparison with these baselines in Section 6.4 of the revised PDF. In the final version of the paper, we are willing to provide comparisons on all tasks from PINNacle.

---

### Author Response · Authors · 2025-12-03

Dear Area Chair,

We understand that the discussion process this year has been unusually challenging for all parties involved. Below, we provide a concise summary of our interactions with the reviewers.

Reviewers Zu1m and Me5S were **satisfied** with our responses and indicated that they no longer have concerns regarding our work.

The discussion with Reviewer akXc was interrupted due to the circumstances, although they also stated that most of their concerns had been **resolved**. Regarding their last remaining question, we have conducted additional experiments, which is included in Appendix D of the revised PDF.

Unfortunately, our discussion with Reviewer xUeL was also left incomplete. They raised several insightful questions, which we believe we have **addressed thoroughly**.

We would also kindly ask the Area Chair to consider the additional experiments presented in Section 6 and Appendices B–D.

We truly appreciate the Area Chair’s efforts and the additional workload they have been required to take on under these circumstances.

Sincerely,

the Authors.

---

### Meta-Review · Area_Chair_TNLp · 2026-01-07

**Summary:**

This paper addresses the instability of Physics-Informed Neural Networks (PINNs) by reformulating the training objective as a nonconvex–strongly concave min–max (saddle-point) problem with a Bregman regularizer to stabilize the weights. Theoretical guarantees are provided, and experimental results on many PDEs are shown.

Fortunately, much discussion remains for this paper. Three of the four reviewers acknowledged that their concerns have mostly been addressed. The last reviewer remains negative, noting that the modest improvements shown here may not be sufficient for acceptance at ICLR. Overall, the average score should likely end up around 6 or slightly below; however, considering that the remaining concerns are not significant, I believe that this paper can be accepted.

**Reviewer Concerns:**

The main concerns include a lack of intuitive theoretical explanations, reproducibility, a lack of experimental details and insufficient improvement in the experiments. These concerns are mostly considered addressed, but one reviewer commented that the magnitude of improvement reported in the experiments is insufficient. It is unclear whether this concern has been addressed or not. However, this concern may not be a sufficient reason to reject the paper.

**Reviewer Scores:**

The scores were 6-4-4-4, but the score should have become 6-6-6-4 since the concerns of the two negative reviewers were resolved during the discussion. If the concerns of the last reviewer are addressed, then the scores would be 6-6-6-6. So the expected average score is 6 or slightly less than that.

---

### Decision · Program_Chairs · 2026-01-26

Accept (Poster)